# Learning with Complementary Labels Revisited: A Consistent Approach via Negative-Unlabeled Learning

## Abstract

Complementary-label learning is a weakly supervised learning problem in which each training example is associated with one or multiple complementary labels indicating the classes to which it does not belong. Existing consistent approaches have relied on the uniform distribution assumption to model the generation of complementary labels, or on an ordinary-label training set to estimate the transition matrix. However, both conditions may not be satisfied in real-world scenarios. In this paper, we propose a novel complementary-label learning approach that does not rely on these conditions. We find that complementary-label learning can be expressed as a set of negative-unlabeled binary classification problems when using the one-versus-rest strategy. This observation allows us to propose a risk-consistent approach with theoretical guarantees. Furthermore, we introduce a risk correction approach to address overfitting problems when using complex models. We also prove the statistical consistency and convergence rate of the corrected risk estimator. Extensive experimental results on both synthetic and real-world benchmark datasets validate the superiority of our proposed approach over state-of-the-art methods.

## 1 Introduction

Deep learning and its applications have achieved great success in recent years. However, to achieve good performance, large amounts of training data with accurate labels are required, which may not be satisfied in some real-world scenarios. Due to the effectiveness in reducing the cost and effort of labeling while maintaining comparable performance, various weakly supervised learning problems have been investigated in recent years, including semi-supervised learning (Berthelot et al., 2019), noisy-label learning (Patrini et al., 2017), programmatic weak supervision (Zhang et al., 2021a), positive-unlabeled learning (Bekker & Davis, 2020), similarity-based classification (Hsu et al., 2019), and partial-label learning (Wang et al., 2022).

Complementary-label learning is another weakly supervised learning problem that has received a lot of attention recently (Ishida et al., 2017). In complementary-label learning, we are given training data associated with complementary labels that specify the classes to which the examples do not belong. The task is to learn a multi-class classifier that assigns correct labels to ordinary-label testing data. Collecting training data with complementary labels is much easier and cheaper than collecting ordinary-label data. For example, when asking workers on crowdsourcing platforms to annotate training data, we only need to randomly select a candidate label and then ask them whether the example belongs to that class or not. Such "yes" or "no" questions are much easier to answer than asking workers to determine the ground-truth label from candidate labels. The benefits and effectiveness of complementary-label learning have also been demonstrated in several machine learning problems and applications, such as domain adaptation (Han et al., 2023; Zhang et al., 2021b), semi-supervised learning (Chen et al., 2020b; Deng et al., 2022), noisy-label learning (Kim et al., 2019), adversarial robustness (Zhou et al., 2022), and medical image analysis (Rezaei et al., 2020).

Existing research works with *consistency guarantees* have attempted to solve complementary-label learning problems by making assumptions about the distribution of complementary labels. The remedy started with Ishida et al. (2017), which proposed the *uniform distribution assumption* that

Table 1: Comparison between CONU and previous risk-consistent or classifier-consistent methods.

| Method | Uniform distribution assumption-free | Ordinary-label training set-free | Classifier-consistent | Risk-consistent |
|---|---|---|---|---|
| PC (Ishida et al., 2017) | ✗ | ✓ | ✓ | ✓ |
| Forward (Yu et al., 2018) | ✓ | ✗ | ✓ | ✗ |
| NN (Ishida et al., 2019) | ✗ | ✓ | ✓ | ✓ |
| LMCL (Feng et al., 2020a) | ✗ | ✓ | ✓ | ✓ |
| OP (Liu et al., 2023) | ✗ | ✓ | ✓ | ✗ |
| CONU | ✓ | ✓ | ✓ | ✓[1] |

a label other than the ground-truth label is sampled from the uniform distribution to be the complementary label. A subsequent work extended it to arbitrary loss functions and models (Ishida et al., 2019) based on the same distribution assumption. Then, Feng et al. (2020a) extended the problem setting to the existence of multiple complementary labels. Recent works have proposed discriminative methods that work by modelling the posterior probabilities of complementary labels instead of the generation process (Chou et al., 2020; Gao & Zhang, 2021; Liu et al., 2023; Lin & Lin, 2023). However, the uniform distribution assumption is still necessary to ensure the classifier consistency property (Liu et al., 2023). Yu et al. (2018) proposed the *biased distribution assumption*, elaborating that the generation of complementary labels follows a *transition matrix*, i.e., the complementary-label distribution is determined by the true label.

In summary, previous complementary-label learning approaches all require either the uniform distribution assumption or the biased distribution assumption to guarantee the consistency property, to the best of our knowledge. However, such assumptions may not be satisfied in real-world scenarios. On the one hand, the uniform distribution assumption is too strong, since the transition probability for different complementary labels is undifferentiated, i.e., the transition probability from the true label to a complementary label is constant for all labels. Such an assumption is not realistic since the annotations may be imbalanced and biased (Wei et al., 2023; Wang et al., 2023). On the other hand, although the biased distribution assumption is more practical, an ordinary-label training set with *deterministic labels*, also known as *anchor points* (Liu & Tao, 2015), is essential for estimating transition probabilities during the training phase (Yu et al., 2018). However, the collection of ordinary-label data with deterministic labels is often unrealistic in complementary-label learning problems (Feng et al., 2020a; Gao & Zhang, 2021).

To this end, we propose a novel risk-consistent approach named CONU, i.e., *COmplementary-label learning via Negative-Unlabeled learning*, without relying on the uniform distribution assumption or an additional ordinary-label training set. Based on an assumption milder than the uniform distribution assumption, we show that the complementary-label learning problem can be equivalently expressed as a set of negative-unlabeled binary classification problems based on the one-versus-rest strategy. Then, a risk-consistent method is deduced with theoretical guarantees. Table 1 shows the comparison between CONU and previous methods. The main contributions of this work are summarized as follows:

- Methodologically, we propose the first consistent complementary-label learning approach without relying on the uniform distribution assumption or an additional ordinary-label dataset.
- Theoretically, we uncover the relationship between complementary-label learning and negative-unlabeled learning, which provides a new perspective for understanding complementary-label learning. The consistency and convergence rate of the corrected risk estimator are proved.
- Empirically, the proposed approach is shown to achieve superior performance over state-of-the-art methods on both synthetic and real-world benchmark datasets.

## 2 PRELIMINARIES

In this section, we introduce the notations used in this paper and briefly discuss the background of ordinary multi-class classification and positive-unlabeled learning.

---

[1]The risk consistency is w.r.t. the one-versus-rest risk.

## 2.1 MULTI-CLASS CLASSIFICATION

Let $\mathcal{X} = \mathbb{R}^d$ denote the $d$-dimensional feature space and $\mathcal{Y} = \{1, 2, \ldots, q\}$ denote the label space with $q$ class labels. Let $p(\boldsymbol{x}, y)$ be the joint probability density over the random variables $(\boldsymbol{x}, y) \in \mathcal{X} \times \mathcal{Y}$. Let $\pi_k = p(y = k)$ be the class-prior probability of the $k$-th class and $p(\boldsymbol{x}|y = k)$ denote the class-conditional density. Besides, let $p(\boldsymbol{x})$ denote the marginal density of unlabeled data. Then, an ordinary-label dataset $\mathcal{D}^{\mathrm{O}} = \{(\boldsymbol{x}_i, y_i)\}_{i=1}^n$ consists of $n$ training examples sampled independently from $p(\boldsymbol{x}, y)$. In this paper, we consider the *one-versus-rest (OVR)* strategy to solve multi-class classification, which is a common strategy with extensive theoretical guarantees and sound performance (Rifkin & Klautau, 2004; Zhang, 2004). Accordingly, the classification risk is

$$R(f_1, f_2, \ldots, f_q) = \mathbb{E}_{p(\boldsymbol{x}, y)} \left[ \ell(f_y(\boldsymbol{x})) + \sum_{k \in \mathcal{Y} \setminus \{y\}} \ell(-f_k(\boldsymbol{x})) \right]. \tag{1}$$

Here, $f_k$ is a binary classifier w.r.t. the $k$-th class, $\mathbb{E}$ denotes the expectation, and $\ell : \mathbb{R} \to \mathbb{R}_+$ is a non-negative binary-class loss function. Then, the predicted label for a testing instance $\boldsymbol{x}$ is determined as $f(\boldsymbol{x}) = \arg\max_{k \in \mathcal{Y}} f_k(\boldsymbol{x})$. The goal is to find optimal classifiers $f_1^*, f_2^*, \ldots, f_q^*$ in a function class $\mathcal{F}$ which achieve the minimum classification risk in Eq. (1), i.e., $(f_1^*, f_2^*, \ldots, f_q^*) = \arg\min_{f_1, f_2, \ldots, f_q \in \mathcal{F}} R(f_1, f_2, \ldots, f_q)$. However, since the joint probability distribution is unknown in practice, the classification risk in Eq. (1) is often approximated by the empirical risk $\widehat{R}(f_1, f_2, \ldots, f_q) = \sum_{i=1}^n \left( \ell(f_{y_i}(\boldsymbol{x}_i)) + \sum_{k \in \mathcal{Y} \setminus \{y_i\}} \ell(-f_k(\boldsymbol{x}_i)) \right) / n$. Accordingly, the optimal classifier w.r.t. the empirical risk is $(\widehat{f}_1, \widehat{f}_2, \ldots, \widehat{f}_q) = \arg\min_{f_1, f_2, \ldots, f_q \in \mathcal{F}} \widehat{R}(f_1, f_2, \ldots, f_q)$. We may add regularization terms to $\widehat{R}(f_1, f_2, \ldots, f_q)$ when necessary (Loshchilov & Hutter, 2019). Also, when using deep neural networks as the backbone model, we typically share the representation layers and only use different classification layers for different labels (Wen et al., 2021).

## 2.2 POSITIVE-UNLABELED LEARNING

In positive-unlabeled (PU) learning (Elkan & Noto, 2008; du Plessis et al., 2014; Kiryo et al., 2017), the goal is to learn a binary classifier only from a positive dataset $\mathcal{D}^{\mathrm{P}} = \{(\boldsymbol{x}_i, +1)\}_{i=1}^{n^{\mathrm{P}}}$ and an unlabeled dataset $\mathcal{D}^{\mathrm{U}} = \{\boldsymbol{x}_i\}_{i=1}^{n^{\mathrm{U}}}$. Based on different assumptions about the data generation process, there are mainly two problem settings for PU learning, i.e., the case-control setting (du Plessis et al., 2014; Niu et al., 2016) and the single-training-set setting (Elkan & Noto, 2008). In the case-control setting, we assume that $\mathcal{D}^{\mathrm{P}}$ is sampled from the positive-class density $p(\boldsymbol{x}|y = +1)$ and $\mathcal{D}^{\mathrm{U}}$ is sampled from the marginal density $p(\boldsymbol{x})$. In contrast, in the single-training-set setting, we assume that an unlabeled dataset is first sampled from the marginal density $p(\boldsymbol{x})$. Then, if a training example is positive, its label is observed with a *constant probability $c$*, and the example remains unlabeled with probability $1 - c$. If a training example is negative, its label is never observed and the example remains unlabeled with probability 1. In this paper, we make use of the single-training-set setting.

## 3 METHODOLOGY

In this section, we first elaborate the generation process of complementary labels. Then, we present a novel unbiased risk estimator (URE) for complementary-label learning with extensive theoretical analysis. Furthermore, a risk correction approach is introduced to improve the generalization performance with risk consistency guarantees.

## 3.1 DATA GENERATION PROCESS

In complementary-label learning, each training example is associated with one or multiple complementary labels specifying the classes to which the example does not belong. Let $\mathcal{D} = \{(\boldsymbol{x}_i, \bar{Y}_i)\}_{i=1}^n$ denote the complementary-label training set sampled i.i.d. from an unknown distribution $p(\boldsymbol{x}, \bar{Y})$. Here, $\boldsymbol{x} \in \mathcal{X}$ is a feature vector, and $\bar{Y} \subseteq \mathcal{Y}$ is a complementary-label set associated with $\boldsymbol{x}$. Traditional complementary-label learning problems can be categorized into single complementary-label

learning (Ishida et al., 2017; Gao & Zhang, 2021; Liu et al., 2023) and multiple complementary-label learning (Feng et al., 2020a). In this paper, we consider a more general case where $\bar{Y}$ may contain *any number* of complementary labels, ranging from zero to $q - 1$. For ease of notation, we use a $q$-dimensional label vector $\bar{\boldsymbol{y}} = [\bar{y}_1, \bar{y}_2, \ldots, \bar{y}_q] \in \{0, 1\}^q$ to denote the vector version of $\bar{Y}$, where $\bar{y}_k = 1$ when $k \in \bar{Y}$ and $\bar{y}_k = 0$ otherwise. Let $\bar{\pi}_k = p(\bar{y}_k = 1)$ denote the fraction of training data where the $k$-th class is considered as a complementary label. Let $p(\boldsymbol{x}|\bar{y}_k = 1)$ and $p(\boldsymbol{x}|\bar{y}_k = 0)$ denote the marginal densities where the $k$-th class is considered as a complementary label or not. The task of complementary-label learning is to learn a multi-class classifier $f : \mathcal{X} \to \mathcal{Y}$ from $\mathcal{D}$.

Inspired by the Selected Completely At Random (SCAR) assumption in PU learning (Elkan & Noto, 2008; Coudray et al., 2023), we propose the SCAR assumption for generating complementary labels, which can be summarized as follows.

**Assumption 1** (Selected Completely At Random (SCAR)). The training examples with the $k$-th class as a complementary label are sampled completely at random from the marginal density of data not belonging to the $k$-th class, i.e.,

$$p(k \in \bar{Y}|\boldsymbol{x}, k \in \mathcal{Y}\backslash\{y\}) = p(k \in \bar{Y}|k \in \mathcal{Y}\backslash\{y\}) = c_k = \bar{\pi}_k/(1 - \pi_k), \qquad (2)$$

where $c_k$ is a constant related to the $k$-th class.

The above assumption means that the sampling procedure is independent of the features and ground-truth labels. Notably, such an assumption is milder than the uniform distribution assumption because the transition probabilities can be different for different complementary labels. Besides, our assumption differs from the biased distribution assumption in that we do not require the transition matrix to be normalized in the row. Based on the SCAR assumption, we generate the complementary-label training set $\mathcal{D}$ as follows. First, an unlabeled dataset is sampled from $p(\boldsymbol{x})$. Then, if the latent ground-truth label of an example is not the $k$-th class, we assign it a complementary label $k$ with probability $c_k$ and still consider it to be an unlabeled example with probability $1 - c_k$. We generate complementary labels for all the examples by following the procedure w.r.t. each of the $q$ labels.

### 3.2 UNBIASED RISK ESTIMATOR

First, we show that the ordinary multi-class classification risk in Eq. (1) can be expressed using examples sampled from $p(\boldsymbol{x}|\bar{y}_k = 1)$ and $p(\boldsymbol{x}|\bar{y}_k = 0)$ (the proof is given in Appendix C).

**Theorem 1.** *Based on Assumption 1, the classification risk in Eq. (1) can be equivalently expressed as* $R(f_1, f_2, \ldots, f_q) = \sum_{k=1}^q R_k(f_k)$, *where*

$$R_k(f_k) = \mathbb{E}_{p(\boldsymbol{x}|\bar{y}_k=1)}\left[(1 - \pi_k)\ell(-f_k(\boldsymbol{x})) + (\bar{\pi}_k + \pi_k - 1)\ell(f_k(\boldsymbol{x}))\right]$$
$$+ \mathbb{E}_{p(\boldsymbol{x}|\bar{y}_k=0)}\left[(1 - \bar{\pi}_k)\ell(f_k(\boldsymbol{x}))\right].$$

**Remark 1.** We find that the multi-class classification risk in Theorem 1 is the sum of the classification risk in negative-unlabeled learning (Elkan & Noto, 2008) by regarding each class as the positive class in turn. Actually, the proposed approach can be considered as a *general framework* for solving complementary-label learning problems. Apart from minimizing $R_k(f_k)$, we can adopt any other PU learning approach (Chen et al., 2020a; Garg et al., 2021; Li et al., 2022; Wilton et al., 2022) to derive the binary classifier $f_k$ by interchanging the positive class and the negative class. Then, we adopt the OVR strategy to determine the predicted label for testing data.

Since the true densities $p(\boldsymbol{x}|\bar{y}_k = 1)$ and $p(\boldsymbol{x}|\bar{y}_k = 0)$ are not directly accessible, we approximate the risk *empirically*. To this end, we need to collect datasets $\mathcal{D}_k^{\mathrm{N}}$ and $\mathcal{D}_k^{\mathrm{U}}$ sampled i.i.d. from $p(\boldsymbol{x}|\bar{y}_k = 1)$ and $p(\boldsymbol{x}|\bar{y}_k = 0)$, respectively. This paper considers generating these datasets by *duplicating* instances of $\mathcal{D}$. Specifically, if the $k$-th class is a complementary label of a training example, we regard its duplicated instance as a *negative example* sampled from $p(\boldsymbol{x}|\bar{y}_k = 1)$ and put the duplicated instance in $\mathcal{D}_k^{\mathrm{N}}$. If the $k$-th class is not a complementary label of a training example, we regard its duplicated instance as an *unlabeled example* sampled from $p(\boldsymbol{x}|\bar{y}_k = 0)$ and put the duplicated instance in $\mathcal{D}_k^{\mathrm{U}}$. In this way, we can obtain $q$ negative binary-class datasets and $q$ unlabeled binary-class datasets:

$$\mathcal{D}_k^{\mathrm{N}} = \left\{(\boldsymbol{x}_{k,i}^{\mathrm{N}}, -1)\right\}_{i=1}^{n_k^{\mathrm{N}}} = \left\{(\boldsymbol{x}_j, -1)|(\boldsymbol{x}_j, \bar{Y}_j) \in \mathcal{D}, k \in \bar{Y}_j\right\}, \quad \text{where } k \in \mathcal{Y}; \qquad (3)$$

$$\mathcal{D}_k^{\mathrm{U}} = \left\{\boldsymbol{x}_{k,i}^{\mathrm{U}}\right\}_{i=1}^{n_k^{\mathrm{U}}} = \left\{\boldsymbol{x}_j|(\boldsymbol{x}_j, \bar{Y}_j) \in \mathcal{D}, k \notin \bar{Y}_j\right\}, \quad \text{where } k \in \mathcal{Y}. \qquad (4)$$

Then, an unbiased risk estimator can be derived from these binary-class datasets to approximate the classification risk in Theorem 1 as $\widehat{R}(f_1, f_2, \ldots, f_q) = \sum_{k=1}^{q} \widehat{R}_k(f_k)$, where

$$
\begin{aligned}
\widehat{R}_k(f_k) = &\frac{1}{n_k^{\mathrm{N}}} \sum_{i=1}^{n_k^{\mathrm{N}}} \left( (1 - \pi_k) \ell \left( -f_k(\boldsymbol{x}_{k,i}^{\mathrm{N}}) \right) + (\bar{\pi}_k + \pi_k - 1) \ell \left( f_k(\boldsymbol{x}_{k,i}^{\mathrm{N}}) \right) \right) \\
&+ \frac{(1 - \bar{\pi}_k)}{n_k^{\mathrm{U}}} \sum_{i=1}^{n_k^{\mathrm{U}}} \ell \left( f_k(\boldsymbol{x}_{k,i}^{\mathrm{U}}) \right).
\end{aligned}
\tag{5}
$$

When the class priors $\pi_k$ are not accessible to the learning algorithm, they can be estimated by off-the-shelf mixture proportion estimation approaches (Ramaswamy et al., 2016; Scott, 2015; Garg et al., 2021; Yao et al., 2022). Notably, the *irreducibility* (Blanchard et al., 2010; Scott et al., 2013) assumption is necessary for class-prior estimation. However, it is still less demanding than the biased distribution assumption, which requires additional ordinary-label training data with deterministic labels, a.k.a. anchor points, to estimate the transition matrix (Yu et al., 2018). Due to page limitations, we present the details of a class-prior estimation algorithm in Appendix A.

## 3.3 Theoretical Analysis

**Infinite-sample consistency.** Since the OVR strategy is used, it remains unknown whether the proposed risk can be calibrated to the 0-1 loss. We answer this question in the affirmative by providing infinite-sample consistency. Let $R_{0-1}(f) = \mathbb{E}_{p(\boldsymbol{x},y)}\mathbb{I}(f(\boldsymbol{x}) \neq y)$ denote the expected 0-1 loss where $f(\boldsymbol{x}) = \arg\max_{k \in \mathcal{Y}} f_k(\boldsymbol{x})$ and $R_{0-1}^* = \min_f R_{0-1}(f)$ denote the Bayes error. Besides, let $R^* = \min_{f_1, f_2, \ldots, f_q} R(f_1, f_2, \ldots, f_q)$ denote the minimum risk of the proposed risk. Then we have the following theorem (its proof is given in Appendix D).

**Theorem 2.** *Suppose the binary-class loss function $\ell$ is convex, bounded below, differential, and satisfies $\ell(z) \leq \ell(-z)$ when $z > 0$. Then we have that for any $\epsilon_1 > 0$, there exists a $\epsilon_2 > 0$ such that*

$$
R(f_1, f_2, \ldots, f_q) \leq R^* + \epsilon_2 \Rightarrow R_{0-1}(f) \leq R_{0-1}^* + \epsilon_1.
\tag{6}
$$

**Remark 2.** The infinite-sample consistency elucidates that the proposed risk can be calibrated to the 0-1 loss. Therefore, if we minimize the risk and obtain the optimal classifier, the classifier also achieves the Bayes error.

**Estimation error bound.** We further elaborate the convergence property of the empirical risk estimator $\widehat{R}(f_1, f_2, \ldots, f_q)$ by providing its estimation error bound. We assume that there exists some constant $C_f$ such that $\sup_{f \in \mathcal{F}} \|f\|_\infty \leq C_f$ and some constant $C_\ell$ such that $\sup_{|z| \leq C_f} \ell(z) \leq C_\ell$. We also assume that the binary-class loss function $\ell(z)$ is Lipschitz continuous w.r.t. $z$ with a Lipschitz constant $L_\ell$. Then we have the following theorem (its proof is given in Appendix E).

**Theorem 3.** *Based on the above assumptions, for any $\delta > 0$, the following inequality holds with probability at least $1 - \delta$:*

$$
R\left(\widehat{f}_1, \widehat{f}_2, \ldots, \widehat{f}_q\right) - R\left(f_1^*, f_2^*, \ldots, f_q^*\right) \leq \sum_{k=1}^{q} \left( (1 - \bar{\pi}_k) C_\ell \sqrt{\frac{2\ln(2/\delta)}{n_k^{\mathrm{U}}}} + (4 - 4\bar{\pi}_k) L_\ell \mathfrak{R}_{n_k^{\mathrm{U}}, p_k^{\mathrm{U}}}(\mathcal{F}) \right.
$$

$$
\left. + (8 - 8\pi_k - 4\bar{\pi}_k) L_\ell \mathfrak{R}_{n_k^{\mathrm{N}}, p_k^{\mathrm{N}}}(\mathcal{F}) + (2 - 2\pi_k - \bar{\pi}_k) C_\ell \sqrt{\frac{2\ln(2/\delta)}{n_k^{\mathrm{N}}}} \right).
\tag{7}
$$

*where $\mathfrak{R}_{n_k^{\mathrm{U}}, p_k^{\mathrm{U}}}(\mathcal{F})$ and $\mathfrak{R}_{n_k^{\mathrm{N}}, p_k^{\mathrm{N}}}(\mathcal{F})$ denote the Rademacher complexity of $\mathcal{F}$ given $n_k^{\mathrm{U}}$ unlabeled data sampled from $p(\boldsymbol{x}|\bar{y}_k = 0)$ and $n_k^{\mathrm{N}}$ negative data sampled from $p(\boldsymbol{x}|\bar{y}_k = 1)$ respectively.*

**Remark 3.** Theorem 3 elucidates an estimation error bound of our proposed risk estimator. When $n_k^{\mathrm{U}}$ and $n_k^{\mathrm{N}} \to \infty$, $R\left(\widehat{f}_1, \widehat{f}_2, \ldots, \widehat{f}_q\right) \to R\left(f_1^*, f_2^*, \ldots, f_q^*\right)$ because $\mathfrak{R}_{n_k^{\mathrm{U}}, p_k^{\mathrm{U}}}(\mathcal{F}) \to 0$ and $\mathfrak{R}_{n_k^{\mathrm{N}}, p_k^{\mathrm{N}}}(\mathcal{F}) \to 0$ for all parametric models with a bounded norm such as deep neural networks with weight decay (Golowich et al., 2018). Furthermore, the estimation error bound converges in $\mathcal{O}_p\left(\sum_{k=1}^{q}\left(1/\sqrt{n_k^{\mathrm{N}}} + 1/\sqrt{n_k^{\mathrm{U}}}\right)\right)$, where $\mathcal{O}_p$ denotes the order in probability.

### 3.4 Risk Correction Approach

Although the URE has sound theoretical properties, we have found that it can encounter several overfitting problems when using complex models such as deep neural networks. The training curves and testing curves of the method that works by minimizing the URE in Eq. (5) are shown in Figure 1. We refer to the method that works by minimizing the corrected risk estimator in Eq. (9) introduced below as CONU, where the algorithm details are summarized in Appendix B. We can observe that

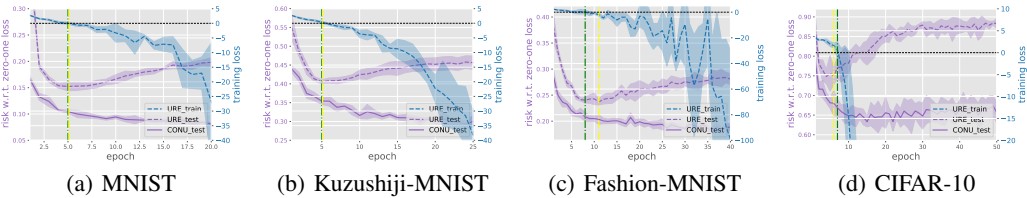

| (a) MNIST | (b) Kuzushiji-MNIST | (c) Fashion-MNIST | (d) CIFAR-10 |

Figure 1: Training curves and testing curves of the method that minimizes the URE and testing curves of our proposed risk correction approach CONU. The green dashed lines indicate when the URE becomes negative while the yellow dashed lines indicate when the overfitting phenomena occur. The complementary labels are generated by following the uniform distribution assumption. ResNet is used as the model architecture for CIFAR-10 and MLP is used for other datasets.

the overfitting phenomena often occur almost simultaneously when the training loss becomes negative. We conjecture the overfitting problems arise from the negative terms in Eq. (5) (Cao et al., 2021; Kiryo et al., 2017; Lu et al., 2020). Therefore, following Ishida et al. (2019); Kiryo et al. (2017), we wrap each potentially negative term with a *non-negative risk correction function* $g(z)$, such as the absolute value function $g(z) = |z|$. For ease of notation, we introduce

$$\widehat{R}_k^{\mathrm{P}}(f_k) = \frac{\bar{\pi}_k + \pi_k - 1}{n_k^{\mathrm{N}}} \sum_{i=1}^{n_k^{\mathrm{N}}} \ell\left(f_k(\boldsymbol{x}_{k,i}^{\mathrm{N}})\right) + \frac{1 - \bar{\pi}_k}{n_k^{\mathrm{U}}} \sum_{i=1}^{n_k^{\mathrm{U}}} \ell\left(f_k(\boldsymbol{x}_{k,i}^{\mathrm{U}})\right). \tag{8}$$

Then, the corrected risk estimator can be written as $\widetilde{R}(f_1, f_2, \ldots, f_q) = \sum_{k=1}^{q} \widetilde{R}_k(f_k)$, where

$$\widetilde{R}_k(f_k) = g\left(\widehat{R}_k^{\mathrm{P}}(f_k)\right) + \frac{1 - \pi_k}{n_k^{\mathrm{N}}} \sum_{i=1}^{n_k^{\mathrm{N}}} \ell\left(-f_k(\boldsymbol{x}_{k,i}^{\mathrm{N}})\right). \tag{9}$$

It is obvious that Eq. (9) is an upper bound of Eq. (5), so the bias is always present. Therefore, it remains doubtful whether the corrected risk estimator is still risk-consistent. Next, we perform a theoretical analysis to clarify that the corrected risk estimator is *biased but consistent*. Since $\mathbb{E}\left[\widehat{R}_k^{\mathrm{P}}(f_k)\right] = \pi_k \mathbb{E}_{p(\boldsymbol{x}|y=k)} \ell(f_k(\boldsymbol{x}))$ is non-negative, we assume that there exists a *non-negative constant* $\beta$ such that for $\forall k \in \mathcal{Y}, \mathbb{E}\left[\widehat{R}_k^{\mathrm{P}}(f_k)\right] \geq \beta$. Besides, we assume that the risk correction function $g(z)$ is Lipschitz continuous with a Lipschitz constant $L_g$. We also assume that the assumptions of Theorem 3 still hold. Let $\left(\widetilde{f}_1, \widetilde{f}_2, \ldots, \widetilde{f}_q\right) = \arg\min_{f_1, f_2, \ldots, f_q \in \mathcal{F}} \widetilde{R}(f_1, f_2, \ldots, f_q)$, then we have the following theorems (the proofs are given in Appendix F and G respectively).

**Theorem 4.** *Based on the above assumptions, the bias of the expectation of the corrected risk estimator has the following lower and upper bounds:*

$$0 \leq \mathbb{E}[\widetilde{R}(f_1, f_2, \ldots, f_q)] - R(f_1, f_2, \ldots, f_q) \leq \sum_{k=1}^{q} (2 - 2\bar{\pi}_k - \pi_k)(L_g + 1) C_\ell \Delta_k, \tag{10}$$

*where* $\Delta_k = \exp\left(-2\beta^2 / \left((1 - \pi_k - \bar{\pi}_k)^2 C_\ell^2 / n_k^{\mathrm{N}} + (1 - \bar{\pi}_k)^2 C_\ell^2 / n_k^{\mathrm{U}}\right)\right)$. *Furthermore, for any* $\delta > 0$*, the following inequality holds with probability at least* $1 - \delta$*:*

$$|\widetilde{R}(f_1, f_2, \ldots, f_q) - R(f_1, f_2, \ldots, f_q)| \leq \sum_{k=1}^{q} \left((1 - \bar{\pi}_k) C_\ell L_g \sqrt{\frac{\ln(2/\delta)}{2n_k^{\mathrm{U}}}}\right.$$

$$\left. + (2 - 2\bar{\pi}_k - \pi_k)(L_g + 1) C_\ell \Delta_k + ((1 - \pi_k - \bar{\pi}_k) L_g + 1 - \pi_k) C_\ell \sqrt{\frac{\ln(2/\delta)}{2n_k^{\mathrm{N}}}}\right).$$

**Theorem 5.** *Based on the above assumptions, for any $\delta > 0$, the following inequality holds with probability at least $1 - \delta$:*

$$R(\widetilde{f}_1, \widetilde{f}_2, \ldots, \widetilde{f}_q) - R(f_1^*, f_2^*, \ldots, f_q^*) \leq \sum_{k=1}^{q} (4 - 4\bar{\pi}_k - 2\pi_k)(L_g + 1) C_\ell \Delta_k$$

$$+ \sum_{k=1}^{q} \left( (1 - \bar{\pi}_k) C_\ell L_g \sqrt{\frac{2\ln(1/\delta)}{n_k^{\mathrm{U}}}} + ((1 - \pi_k - \bar{\pi}_k) L_g + 1 - \pi_k) C_\ell \sqrt{\frac{2\ln(1/\delta)}{n_k^{\mathrm{N}}}} \right)$$

$$+ \sum_{k=1}^{q} \left( (8 - 8\bar{\pi}_k) L_g L_\ell \mathfrak{R}_{n_k^{\mathrm{U}}, p_k^{\mathrm{U}}}(\mathcal{F}) + ((8 - 8\pi_k - 8\bar{\pi}_k) L_g + 8 - 8\pi_k) L_\ell \mathfrak{R}_{n_k^{\mathrm{N}}, p_k^{\mathrm{N}}}(\mathcal{F}) \right).$$

**Remark 4.** Theorem 4 elaborates that $\widetilde{R}(f_1, f_2, \ldots, f_q) \to R(f_1, f_2, \ldots, f_q)$ as $n \to \infty$, which indicates that the corrected risk estimator is biased but consistent. An estimation error bound is also shown in Theorem 5. For $n \to \infty$, $R(\widetilde{f}_1, \widetilde{f}_2, \ldots, \widetilde{f}_q) \to R(f_1^*, f_2^*, \ldots, f_q^*)$ because $\Delta_k \to 0$, $\mathfrak{R}_{n, p^{\mathrm{U}}} \to 0$, and $\mathfrak{R}_{n_k^{\mathrm{N}}, p_k^{\mathrm{N}}} \to 0$ for all parametric models with a bounded norm (Mohri et al., 2012). The convergence rate of the estimation error bound is still $\mathcal{O}_p\left( \sum_{k=1}^{q} \left( 1/\sqrt{n_k^{\mathrm{N}}} + 1/\sqrt{n_k^{\mathrm{U}}} \right) \right)$.

## 4 EXPERIMENTS

In this section, we validate the effectiveness of CONU through extensive experiments.

### 4.1 EXPERIMENTS ON SYNTHETIC BENCHMARK DATASETS

We conducted experiments on synthetic benchmark datasets, including MNIST (LeCun et al., 1998), Kuzushiji-MNIST (Clanuwat et al., 2018), Fashion-MNIST (Xiao et al., 2017), and CIFAR-10 (Krizhevsky & Hinton, 2009). We considered various generation processes of complementary labels by following the uniform, biased, and SCAR assumptions. Details of the datasets, models, and hyperparameters can be found in Appendix H. We considered the single complementary-label setting and similar results could be observed with multiple complementary labels. We evaluated the classification performance of CONU against six single complementary-label learning methods, including PC (Ishida et al., 2017), NN (Ishida et al., 2019), GA (Ishida et al., 2019), L-UW (Gao & Zhang, 2021), L-W (Gao & Zhang, 2021), and OP (Liu et al., 2023). We assumed that the class priors were accessible to the learning algorithm. Since it is not easy to tune hyperparameters for complementary-label learning approaches without an additional ordinary-label dataset (Wang et al., 2023), we adopted the same hyperparameter settings for all the compared approaches for a fair comparison. We randomly generated complementary labels five times with different seeds and recorded the mean accuracy and standard deviations. In addition, a pairwise *t*-test at the 0.05 significance level is performed to show whether the performance advantages are significant.

Tables 2, 3, and 4 show the classification performance of each method with different models and generation settings of complementary labels on MNIST, Kuzushiji-MNIST, and Fashion-MNIST respectively. The experimental results on CIFAR-10 are shown in Appendix J. It is surprising to observe that CONU has achieved the best performance in almost all the settings with different model architectures. This clearly illustrates the superiority of our method in tackling different types of complementary labels.

### 4.2 EXPERIMENTS ON REAL-WORLD BENCHMARK DATASETS

We also verified the effectiveness of CONU on two real-world complementary-label datasets CLCIFAR-10 and CLCIFAR-20 (Wang et al., 2023). The datasets were annotated by human annotators from Amazon Mechanical Turk (MTurk). The distribution of complementary labels is too complex to be captured by any of the above assumptions. Moreover, the complementary labels may be *noisy*, which means that the complementary labels may be annotated as ground-truth labels by mistake. Therefore, both datasets can be used to test the robustness of methods in more realistic environments. There are three human-annotated complementary labels for each example, so they can be considered as multiple complementary-label datasets. We evaluated the classification performance of CONU against eight multiple complementary-label learning or partial-label learning

Table 2: Classification accuracy (mean±std) of each method on MNIST. The best performance is shown in bold (pairwise *t*-test at the 0.05 significance level).

| Setting | Uniform | | Biased-a | | Biased-b | | SCAR-a | | SCAR-b | |
|---|---|---|---|---|---|---|---|---|---|---|
| Model | MLP | LeNet | MLP | LeNet | MLP | LeNet | MLP | LeNet | MLP | LeNet |
| PC | 71.11 ±0.83 | 82.69 ±1.15 | 69.29 ±0.97 | 87.82 ±0.69 | 71.59 ±0.85 | 87.66 ±0.66 | 66.97 ±1.03 | 11.00 ±0.79 | 57.67 ±0.98 | 49.17 ±35.9 |
| NN | 67.75 ±0.96 | 86.16 ±0.69 | 30.59 ±2.31 | 46.27 ±2.61 | 38.50 ±3.93 | 63.67 ±3.75 | 67.39 ±0.68 | 86.58 ±0.95 | 63.95 ±0.56 | 79.94 ±0.48 |
| GA | 88.00 ±0.85 | 96.02 ±0.15 | 65.97 ±7.87 | 94.55 ±0.43 | 75.77 ±1.48 | 94.87 ±0.28 | 62.62 ±2.29 | 90.23 ±0.92 | 56.91 ±2.08 | 78.66 ±0.61 |
| L-UW | 73.49 ±0.88 | 77.74 ±0.97 | 39.63 ±0.57 | 32.21 ±1.20 | 42.77 ±1.42 | 34.57 ±1.90 | 35.08 ±1.59 | 33.82 ±2.44 | 30.24 ±1.81 | 24.28 ±2.74 |
| L-W | 62.24 ±0.50 | 63.04 ±1.58 | 36.90 ±0.34 | 29.25 ±0.94 | 41.55 ±0.63 | 32.98 ±2.25 | 33.53 ±2.08 | 26.02 ±1.31 | 28.99 ±2.38 | 23.69 ±2.94 |
| OP | 78.87 ±0.46 | 88.76 ±1.68 | 73.46 ±0.71 | 85.96 ±1.02 | 74.16 ±0.52 | 87.23 ±1.31 | 76.29 ±0.23 | 86.94 ±1.94 | 68.12 ±0.51 | 71.67 ±2.30 |
| CONU | **91.27** ±**0.20** | **97.00** ±**0.30** | **88.14** ±**0.70** | **96.14** ±**0.32** | **89.51** ±**0.44** | **96.62** ±**0.10** | **90.98** ±**0.27** | **96.72** ±**0.16** | **81.85** ±**0.25** | **87.05** ±**0.28** |

Table 3: Classification accuracy (mean±std) of each method on Kuzushiji-MNIST. The best performance is shown in bold (pairwise *t*-test at the 0.05 significance level).

| Setting | Uniform | | Biased-a | | Biased-b | | SCAR-a | | SCAR-b | |
|---|---|---|---|---|---|---|---|---|---|---|
| Model | MLP | LeNet | MLP | LeNet | MLP | LeNet | MLP | LeNet | MLP | LeNet |
| PC | 42.93 ±0.33 | 56.79 ±1.54 | 41.60 ±0.97 | 67.39 ±1.04 | 42.53 ±0.80 | 66.81 ±1.33 | 39.58 ±1.35 | 42.59 ±29.8 | 33.95 ±1.14 | 37.67 ±25.3 |
| NN | 39.42 ±0.68 | 58.57 ±1.15 | 23.97 ±2.53 | 31.10 ±2.95 | 29.93 ±1.80 | 48.72 ±2.89 | 39.31 ±1.18 | 56.84 ±2.10 | 38.68 ±0.58 | 56.70 ±1.08 |
| GA | 60.83 ±1.37 | 76.17 ±0.44 | 43.22 ±3.03 | **75.04** ±**0.92** | 48.03 ±2.93 | **77.05** ±**1.67** | 36.56 ±2.96 | 59.16 ±3.30 | 33.02 ±2.31 | 52.92 ±2.39 |
| L-UW | 43.00 ±1.20 | 49.31 ±1.95 | 27.89 ±0.51 | 25.82 ±0.78 | 31.53 ±0.42 | 30.05 ±1.63 | 21.49 ±0.57 | 19.71 ±1.44 | 18.36 ±1.23 | 16.67 ±1.86 |
| L-W | 37.21 ±0.59 | 42.69 ±2.54 | 26.75 ±0.61 | 25.86 ±0.64 | 30.10 ±0.57 | 27.94 ±1.68 | 21.22 ±0.77 | 18.28 ±2.11 | 18.41 ±1.66 | 16.25 ±1.51 |
| OP | 51.78 ±0.41 | 65.94 ±1.38 | 45.66 ±0.90 | 65.59 ±1.71 | 47.47 ±1.26 | 64.65 ±1.68 | 49.95 ±0.79 | 59.93 ±1.38 | 42.72 ±0.95 | 56.36 ±2.15 |
| CONU | **67.95** ±**1.29** | **79.81** ±**1.19** | **62.43** ±**1.02** | 75.99 ±**0.91** | **64.98** ±**0.72** | 78.53 ±**0.57** | **66.72** ±**0.69** | **78.27** ±**1.09** | **61.78** ±**0.36** | **72.03** ±**0.45** |

methods, including CC (Feng et al., 2020b), PRODEN (Lv et al., 2020), EXP (Feng et al., 2020a), MAE (Feng et al., 2020a), Phuber-CE (Feng et al., 2020a), LWS (Wen et al., 2021), CAVL (Zhang et al., 2022), and IDGP (Qiao et al., 2023). We found that the performance of some approaches was unstable with different network initialization, so we randomly initialized the network five times with different seeds and recorded the mean accuracy and standard deviations. Table 5 shows the experimental results on CLCIFAR-10 and CLCIFAR-20 with different models. It is interesting to observe that CONU still achieves the best performance on both datasets, confirming its effectiveness in handling challenging real-world complementary labels.

### 4.3 SENSITIVITY ANALYSIS

In many real-world scenarios, the given or estimated class priors may deviate from the ground-truth values. We investigated the influence of inaccurate class priors on the classification performance of CONU. Specifically, let $\bar{\pi}_k = \epsilon_k \pi_k$ denote the corrupted class prior prob-

Table 4: Classification accuracy (mean±std) of each method on Fashion-MNIST. The best performance is shown in bold (pairwise $t$-test at the 0.05 significance level).

| Setting | Uniform | | Biased-a | | Biased-b | | SCAR-a | | SCAR-b | |
|---|---|---|---|---|---|---|---|---|---|---|
| Model | MLP | LeNet | MLP | LeNet | MLP | LeNet | MLP | LeNet | MLP | LeNet |
| PC | 64.82 ±1.27 | 69.56 ±1.82 | 61.14 ±1.09 | 72.89 ±1.26 | 61.20 ±0.79 | 73.04 ±1.38 | 63.08 ±0.88 | 23.28 ±29.7 | 47.23 ±2.38 | 37.53 ±25.2 |
| NN | 63.89 ±0.92 | 70.34 ±1.09 | 25.66 ±2.12 | 36.93 ±3.86 | 30.75 ±0.96 | 40.88 ±3.71 | 63.47 ±0.70 | 70.83 ±0.87 | 55.96 ±1.55 | 63.06 ±1.38 |
| GA | 77.04 ±0.95 | 81.91 ±0.43 | 50.04 ±4.30 | 74.73 ±0.96 | 49.02 ±5.76 | 75.66 ±1.10 | 54.74 ±3.04 | 74.75 ±1.17 | 44.75 ±3.04 | 60.01 ±1.47 |
| L-UW | **80.29** **±0.44** | 72.43 ±2.07 | 40.26 ±2.49 | 29.46 ±1.70 | 43.55 ±1.61 | 33.53 ±1.35 | 35.71 ±1.50 | 30.73 ±1.64 | 31.43 ±2.98 | 22.03 ±3.62 |
| L-W | 75.14 ±0.40 | 61.89 ±0.88 | 39.87 ±0.95 | 27.57 ±1.70 | 42.02 ±1.41 | 32.69 ±0.68 | 31.86 ±2.16 | 27.37 ±2.30 | 30.26 ±1.68 | 21.61 ±2.12 |
| OP | 69.03 ±0.71 | 71.28 ±0.94 | 62.93 ±1.25 | 70.82 ±1.15 | 62.25 ±0.36 | 68.94 ±2.78 | 66.29 ±0.60 | 69.52 ±1.18 | 56.55 ±1.39 | 56.39 ±3.03 |
| CONU | **80.44** **±0.19** | **82.74** **±0.39** | **70.08** **±2.53** | **79.74** **±1.10** | **71.97** **±1.09** | **80.43** **±0.69** | **79.75** **±0.60** | **82.55** **±0.30** | **71.16** **±0.66** | **72.79** **±0.62** |

Table 5: Classification accuracy (mean±std) of each method on CLCIFAR-10 and CLCIFAR-20. The best performance is shown in bold (pairwise $t$-test at the 0.05 significance level).

| Dataset | Model | CC | PRODEN | EXP | MAE | Phuber-CE | LWS | CAVL | IDGP | CONU |
|---|---|---|---|---|---|---|---|---|---|---|
| CLCIFAR-10 | ResNet | 31.56 ±2.17 | 26.37 ±0.98 | 34.84 ±4.19 | 19.48 ±2.88 | **41.13** **±0.74** | 13.05 ±4.18 | 24.12 ±3.32 | 10.00 ±0.00 | **42.04** **±0.96** |
| | DenseNet | 37.03 ±1.77 | 31.31 ±1.06 | **43.27** **±1.33** | 22.77 ±0.22 | 39.92 ±0.91 | 10.00 ±0.00 | 25.31 ±4.06 | 10.00 ±0.00 | **44.41** **±0.43** |
| CLCIFAR-20 | ResNet | 5.00 ±0.00 | 6.69 ±0.31 | 7.21 ±0.17 | 5.00 ±0.00 | 8.10 ±0.18 | 5.20 ±0.45 | 5.00 ±0.00 | 4.96 ±0.09 | **20.08** **±0.62** |
| | DenseNet | 5.00 ±0.00 | 5.00 ±0.00 | 7.51 ±0.91 | 5.67 ±1.49 | 7.22 ±0.39 | 5.00 ±0.00 | 5.09 ±0.13 | 5.00 ±0.00 | **19.91** **±0.68** |

ability for the $k$-th class where $\epsilon_k$ is sampled from a normal distribution $\mathcal{N}(1, \sigma^2)$. We further normalized the obtained corrupted class priors to ensure that they sum up to one. Figure 2 shows the classification performance given inaccurate class priors on three datasets using the uniform generation process and LeNet as the model architecture. From Figure 2, we can see that the performance is still satisfactory with small perturbations of the class priors. However, the performance will degenerate if the class priors deviate too much from the ground-truth values. Therefore, it is important to obtain accurate class priors to ensure satisfactory performance.

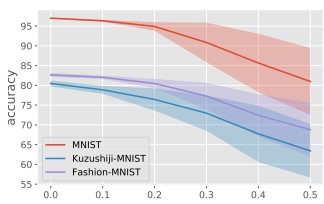

Figure 2: Classification accuracy given inaccurate class priors.

## 5 CONCLUSION

In this paper, we proposed a consistent complementary-label learning approach without relying on the uniform distribution assumption or an ordinary-label training set to estimate the transition matrix. We observed that complementary-label learning could be expressed as a set of negative-unlabeled classification problems based on the OVR strategy. Accordingly, a risk-consistent approach with theoretical guarantees was proposed. Extensive experimental results on benchmark datasets validated the effectiveness of our proposed approach. In the future, it would be interesting to apply the idea to other weakly supervised learning problems.

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

# A  CLASS-PRIOR ESTIMATION

When the class priors $\pi_k$ are not accessible to the learning algorithm, they can be estimated by off-the-shelf mixture proportion estimation approaches (Ramaswamy et al., 2016; Scott, 2015; Garg et al., 2021; Yao et al., 2022). In this section, we discuss the problem formulation and how to adapt a state-of-the-art class-prior estimation method to our problem as an example.

**Mixture proportion estimation.**  Let $F$ be a mixture distribution of two component distributions $G$ and $H$ with a proportion $\theta^*$, i.e.,

$$F = (1 - \theta^*)G + \theta^* H.$$

The task of mixture proportion estimation problems is to estimate $\theta^*$ given training examples sampled from $F$ and $H$. For PU learning, we consider $F = p(\boldsymbol{x})$, $G = p(\boldsymbol{x}|y = -1)$, and $H = p(\boldsymbol{x}|y = +1)$. Then, the estimation of $\theta^*$ corresponds to the estimation of the class prior $p(y = +1)$. It is shown that $\theta^*$ cannot be identified without any additional assumptions (Scott et al., 2013; Scott, 2015). Hence, various assumptions have been proposed to ensure the identifiability, including the irreducibility assumption (Scott et al., 2013), the anchor point assumption (Scott, 2015; Liu & Tao, 2015), the separability assumption (Ramaswamy et al., 2016), etc.

**Best Bin Estimation.**  We use Best Bin Estimation (BBE) (Garg et al., 2021) as the base algorithm for class-prior estimation since it can achieve nice performance with easy implementations. First, they split the PU data into PU training data $\mathcal{D}^{\mathrm{PTr}} = \left\{ \left( \boldsymbol{x}_i^{\mathrm{PTr}}, +1 \right) \right\}_{i=1}^{n^{\mathrm{PTr}}}$ and $\mathcal{D}^{\mathrm{UTr}} = \left\{ \boldsymbol{x}_i^{\mathrm{UTr}} \right\}_{i=1}^{n^{\mathrm{UTr}}}$, and PU validation data $\mathcal{D}^{\mathrm{PVal}} = \left\{ \left( \boldsymbol{x}_i^{\mathrm{PVal}}, +1 \right) \right\}_{i=1}^{n^{\mathrm{PVal}}}$ and $\mathcal{D}^{\mathrm{UVal}} = \left\{ \boldsymbol{x}_i^{\mathrm{UVal}} \right\}_{i=1}^{n^{\mathrm{UVal}}}$. Then, they train a positive-versus-unlabeled (PvU) classifier $f^{\mathrm{PvU}}$ with $\mathcal{D}^{\mathrm{PTr}}$ and $\mathcal{D}^{\mathrm{UTr}}$. They collect the model outputs of PU validation data $\mathcal{Z}^{\mathrm{P}} = \left\{ z_i^{\mathrm{P}} \right\}_{i=1}^{n^{\mathrm{PVal}}}$ and $\mathcal{Z}^{\mathrm{U}} = \left\{ z_i^{\mathrm{U}} \right\}_{i=1}^{n^{\mathrm{UVal}}}$ where $z_i^{\mathrm{P}} = f^{\mathrm{PvU}} \left( \boldsymbol{x}_i^{\mathrm{PVal}} \right)$ and $z_i^{\mathrm{U}} = f^{\mathrm{PvU}} \left( \boldsymbol{x}_i^{\mathrm{UVal}} \right)$. Besides, they introduce $q(z) = \int_{A_z} p(\boldsymbol{x}) \, \mathrm{d}\boldsymbol{x}$ where $A_z = \left\{ \boldsymbol{x} \in \mathcal{X} | f^{\mathrm{PvU}}(\boldsymbol{x}) \geq z \right\}$. Then, $q(z)$ can be regarded as the proportion of data with the model output no less than $z$. For $p(\boldsymbol{x}|y = +1)$ and $p(\boldsymbol{x})$, they define $q^{\mathrm{P}}(z)$ and $q^{\mathrm{U}}(z)$ respectively. they estimate them empirically as

$$\widehat{q}^{\mathrm{P}}(z) = \frac{\sum_{i=1}^{n^{\mathrm{PVal}}} \mathbb{I} \left( f^{\mathrm{PvU}} \left( \boldsymbol{x}_i^{\mathrm{PVal}} \right) \geq z \right)}{n^{\mathrm{PVal}}} \quad \text{and} \quad \widehat{q}^{\mathrm{U}}(z) = \frac{\sum_{i=1}^{n^{\mathrm{UVal}}} \mathbb{I} \left( f^{\mathrm{PvU}} \left( \boldsymbol{x}_i^{\mathrm{UVal}} \right) \geq z \right)}{n^{\mathrm{UVal}}}. \quad (11)$$

Then, they obtain $\widehat{z}$ as

$$\widehat{z} = \underset{z \in [0,1]}{\arg\max} \left( \frac{\widehat{q}^{\mathrm{U}}(z)}{\widehat{q}^{\mathrm{P}}(z)} + \frac{1 + \gamma}{\widehat{q}^{\mathrm{P}}(z)} \left( \sqrt{\frac{\ln(4/\delta)}{2n^{\mathrm{PVal}}}} + \sqrt{\frac{\ln(4/\delta)}{2n^{\mathrm{UVal}}}} \right) \right) \quad (12)$$

where $\gamma$ and $\delta$ are hyperparameters respectively. Finally, they calculate the estimation value of the mixture proportion as

$$\widehat{\theta} = \frac{\widehat{q}^{\mathrm{U}}(\widehat{z})}{\widehat{q}^{\mathrm{P}}(\widehat{z})} \quad (13)$$

and they prove that $\widehat{\theta}$ is an unbiased estimator of $\theta^*$ when satisfying the *pure positive bin assumption*, a variant of the irreducibility assumption. More detailed descriptions of the approach can be found in Garg et al. (2021).

**Class-prior estimation for CONU.**  Our class-prior estimation approach is based on BBE. First, we split complementary-label data into training and validation data. Then, we generate $q$ negative binary-class datasets $\mathcal{D}_k^{\mathrm{NTr}}$ and $q$ unlabeled binary-class datasets $\mathcal{D}_k^{\mathrm{UTr}}$ by Eq. (3) and Eq. (4) with training data ($k \in \mathcal{Y}$). We also generate $q$ negative binary-class datasets $\mathcal{D}_k^{\mathrm{NVal}}$ and $q$ unlabeled binary-class datasets $\mathcal{D}_k^{\mathrm{UVal}}$ by Eq. (3) and Eq. (4) with validation data ($k \in \mathcal{Y}$). Then, we estimate the class priors $1 - \pi_k$ for each label $k \in \mathcal{Y}$ by BBE adapted by interchanging the postive and negative classes. Finally, we normalize $\pi_k$ to ensure that they sum up to one. The algorithm detail is summarized in Algorithm 1.

---

**Algorithm 1** Class-prior Estimation

---

**Input:** Complementary-label training set $\mathcal{D}$.

1: **for** $k \in \mathcal{Y}$ **do**
2:     **Generate** training datasets $\mathcal{D}_k^{\mathrm{NTr}}$, $\mathcal{D}_k^{\mathrm{UTr}}$, validation data $\mathcal{D}_k^{\mathrm{NVal}}$, and $\mathcal{D}_k^{\mathrm{UVal}}$ by Eq. (3) and Eq. (4);
3:     **Estimate** the value of $1 - \pi_k$ by employing the BBE algorithm and interchanging the positive and negative classes;
4: **end for**
5: **Normalize** $\pi_k$ to ensure they sum up to one;

**Output:** Class priors $\pi_k$ ($k \in \mathcal{Y}$).

---

## B   Algorithm Details of CONU

---

**Algorithm 2** CONU

---

**Input:** Complementary-label training set $\mathcal{D}$, class priors $\pi_k$ ($k \in \mathcal{Y}$), unseen instance $\boldsymbol{x}_*$, epoch $T_{\max}$, iteration $I_{\max}$.

1: **for** $t = 1, 2, \ldots, T_{\max}$ **do**
2:     **Shuffle** the complementary-label training set $\mathcal{D}$;
3:     **for** $j = 1, \ldots, I_{\max}$ **do**
4:         **Fetch** mini-batch $\mathcal{D}_j$ from $\mathcal{D}$;
5:         **Update** the shared representation layers and specific classification layers $f_1, f_2, \ldots, f_q$ by minimizing the corrected risk estimator $\widetilde{R}(f_1, f_2, \ldots, f_q)$ in Eq. (9);
6:     **end for**
7: **end for**
8: **Return** $y_* = \arg\max_{k \in \mathcal{Y}} f_k(\boldsymbol{x}_*)$;

**Output:** Predicted label $y_*$.

---

## C   Proof of Theorem 1

First, we introduce the following lemma.

**Lemma 1.** *Based on Assumption 1, we have* $p(\boldsymbol{x}|\bar{y}_k = 1) = p(\boldsymbol{x}|y \neq k)$.

*Proof.* On one hand, we have

$$p(\boldsymbol{x}|\bar{y}_k = 1, y \neq k) = \frac{p(\boldsymbol{x}|\bar{y}_k = 1)\, p(y \neq k|\boldsymbol{x}, \bar{y}_k = 1)}{p(y \neq k|\bar{y}_k = 1)}.$$

According to the definition of complementary labels, we have $p(y \neq k|\boldsymbol{x}, \bar{y}_k = 1) = p(y \neq k|\bar{y}_k = 1) = 1$. Therefore, we have $p(\boldsymbol{x}|\bar{y}_k = 1, y \neq k) = p(\boldsymbol{x}|\bar{y}_k = 1)$. On the other hand, we have

$$p(\boldsymbol{x}|\bar{y}_k = 1, y \neq k) = \frac{p(\boldsymbol{x}|y \neq k)p(\bar{y}_k = 1|\boldsymbol{x}, y \neq k)}{p(\bar{y}_k = 1|y \neq k)} = p(\boldsymbol{x}|y \neq k),$$

where the first equation is derived from Assumption 1. The proof is completed. $\square$

Then, the proof of Theorem 1 is given.

*Proof of Theorem 1.*

$$
\begin{aligned}
R(f_1, f_2, \ldots, f_q) =& \mathbb{E}_{p(\boldsymbol{x},y)} \left[ \ell\left(f_y\left(\boldsymbol{x}\right)\right) + \sum_{k=1, k \neq y}^{q} \ell\left(-f_k\left(\boldsymbol{x}\right)\right) \right] \\
=& \mathbb{E}_{p(\boldsymbol{x},y)} \left[ \sum_{k=1}^{q} \left( \mathbb{I}(k=y)\ell(f_k(\boldsymbol{x})) + \mathbb{I}(k \neq y)\ell(-f_k(\boldsymbol{x})) \right) \right] \\
=& \sum_{k=1}^{q} \mathbb{E}_{p(\boldsymbol{x},y)} \left[ \mathbb{I}(k=y)\ell\left(f_k(\boldsymbol{x})\right) + \mathbb{I}(k \neq y)\ell\left(-f_k(\boldsymbol{x})\right) \right] \\
=& \sum_{k=1}^{q} \left( \pi_k \mathbb{E}_{p(\boldsymbol{x}|y=k)} \left[ \ell\left(f_k(\boldsymbol{x})\right) \right] + (1-\pi_k) \mathbb{E}_{p(\boldsymbol{x}|y \neq k)} \left[ \ell\left(-f_k(\boldsymbol{x})\right) \right] \right) \\
=& \sum_{k=1}^{q} \Big( \mathbb{E}_{p(\boldsymbol{x})} \left[ \ell(f_k(\boldsymbol{x})) \right] - (1-\pi_k) \mathbb{E}_{p(\boldsymbol{x}|y \neq k)} \left[ \ell(f_k(\boldsymbol{x})) \right] \\
& + (1-\pi_k) \mathbb{E}_{p(\boldsymbol{x}|y \neq k)} \left[ \ell(-f_k(\boldsymbol{x})) \right] \Big) \\
=& \sum_{k=1}^{q} \Big( \mathbb{E}_{p(\boldsymbol{x})} \left[ \ell(f_k(\boldsymbol{x})) \right] - (1-\pi_k) \mathbb{E}_{p(\boldsymbol{x}|\bar{y}_k=1)} \left[ \ell(f_k(\boldsymbol{x})) \right] \\
& + (1-\pi_k) \mathbb{E}_{p(\boldsymbol{x}|\bar{y}_k=1)} \left[ \ell(-f_k(\boldsymbol{x})) \right] \Big) \\
=& \sum_{k=1}^{q} \Big( \bar{\pi}_k \mathbb{E}_{p(\boldsymbol{x}|\bar{y}_k=1)} \left[ \ell(f_k(\boldsymbol{x})) \right] + (1-\bar{\pi}_k) \mathbb{E}_{p(\boldsymbol{x}|\bar{y}_k=0)} \left[ \ell(f_k(\boldsymbol{x})) \right] \\
& - (1-\pi_k) \mathbb{E}_{p(\boldsymbol{x}|\bar{y}_k=1)} \left[ \ell(f_k(\boldsymbol{x})) \right] + (1-\pi_k) \mathbb{E}_{p(\boldsymbol{x}|\bar{y}_k=1)} \left[ \ell(-f_k(\boldsymbol{x})) \right] \Big) \\
=& \sum_{k=1}^{q} \Big( \mathbb{E}_{p(\boldsymbol{x}|\bar{y}_k=1)} \left[ (1-\pi_k)\ell\left(-f_k(\boldsymbol{x})\right) + (\bar{\pi}_k + \pi_k - 1)\ell\left(f_k(\boldsymbol{x})\right) \right] \\
& + \mathbb{E}_{p(\boldsymbol{x}|\bar{y}_k=0)} \left[ (1-\bar{\pi}_k)\ell\left(f_k(\boldsymbol{x})\right) \right] \Big).
\end{aligned}
$$

Here, $\mathbb{I}(\cdot)$ returns 1 if predicate holds. Otherwise, $\mathbb{I}(\cdot)$ returns 0. The proof is completed. $\square$

## D  PROOF OF THEOREM 2

To begin with, we show the following theoretical results about infinite-sample consistency from Zhang (2004). For ease of notations, let $\boldsymbol{f}(\boldsymbol{x}) = [f_1(\boldsymbol{x}), f_2(\boldsymbol{x}), \ldots, f_q(\boldsymbol{x})]$ denote the vector form of modeling outputs of all the binary classifiers. First, we elaborate the infinite-sample consistency property of the OVR strategy.

**Theorem 5** (Theorem 10 of Zhang (2004))**.** *Consider the OVR strategy, whose surrogate loss function is defined as $\Psi_y(\boldsymbol{f}(\boldsymbol{x})) = \psi(f_y(\boldsymbol{x})) + \sum_{k \in \mathcal{Y} \setminus \{y\}} \psi(-f_k(\boldsymbol{x}))$. Assume $\psi$ is convex, bounded below, differentiable, and $\psi(z) < \psi(-z)$ when $z > 0$. Then, the OVR strategy is infinite-sample consistent on $\Omega = \mathbb{R}^K$ with respect to 0-1 classification risk.*

Then, we elaborate the relationship between the minimum classification risk of an infinite-sample consistent method and the Bayes error.

**Theorem 6** (Theorem 3 of Zhang (2004))**.** *Let $\mathcal{B}$ be the set of all vector Borel measurable functions, which take values in $\mathbb{R}^q$. For $\Omega \subset \mathbb{R}^q$, let $\mathcal{B}_\Omega = \{\boldsymbol{f} \in \mathcal{B} : \forall \boldsymbol{x}, \boldsymbol{f}(x) \in \Omega\}$. If $[\Psi_y(\cdot)]$ is infinite-sample consistent on $\Omega$ with respect to 0-1 classification risk, then for any $\epsilon_1 > 0$, there exists $\epsilon_2 > 0$ such that for all underlying Borel probability measurable $p$, and $\boldsymbol{f}(\cdot) \in \mathcal{B}_\Omega$,*

$$
\mathbb{E}_{(\boldsymbol{x},y) \sim p}[\Psi_y(\boldsymbol{f}(\boldsymbol{x}))] \leq \inf_{\boldsymbol{f}' \in \mathcal{B}_\Omega} \mathbb{E}_{(\boldsymbol{x},y) \sim p}[\Psi_y(\boldsymbol{f}'(\boldsymbol{x}))] + \epsilon_2 \tag{14}
$$

*implies*

$$
R_{0-1}(T(\boldsymbol{f}(\cdot))) \leq R_{0-1}^* + \epsilon_1, \tag{15}
$$

*where $T(\cdot)$ is defined as $T(\boldsymbol{f}(\boldsymbol{x})) := \arg\max_{k=1,\ldots,q} f_k(\boldsymbol{x})$.*

Then, we give the proof of Theorem 2.

*Proof of Theorem 2.* According to Theorem 1, the proposed classification risk $R(f_1, f_2, \ldots, f_q)$ is equivalent to the OVR risk. Therefore, it is sufficient to elaborate the theoretical properties of the OVR risk to prove Theorem 2. □

## E  PROOF OF THEOREM 3

First, we give the definition of Rademacher complexity.

**Definition 1** (Rademacher complexity). Let $\mathcal{X}_n = \{\boldsymbol{x}_1, \ldots \boldsymbol{x}_n\}$ denote $n$ i.i.d. random variables drawn from a probability distribution with density $p(\boldsymbol{x})$, $\mathcal{F} = \{f : \mathcal{X} \mapsto \mathbb{R}\}$ denote a class of measurable functions, and $\boldsymbol{\sigma} = (\sigma_1, \sigma_2, \ldots, \sigma_n)$ denote Rademacher variables taking values from $\{+1, -1\}$ uniformly. Then, the (expected) Rademacher complexity of $\mathcal{F}$ is defined as

$$\mathfrak{R}_{n,p}(\mathcal{F}) = \mathbb{E}_{\mathcal{X}_n} \mathbb{E}_{\boldsymbol{\sigma}} \left[ \sup_{f \in \mathcal{F}} \frac{1}{n} \sum_{i=1}^{n} \sigma_i f(\boldsymbol{x}_i) \right]. \tag{16}$$

For ease of notation, we define $\bar{\mathcal{D}} = \mathcal{D}_1^{\mathrm{U}} \bigcup \mathcal{D}_2^{\mathrm{U}} \bigcup \ldots \bigcup \mathcal{D}_q^{\mathrm{U}} \bigcup \mathcal{D}_1^{\mathrm{N}} \bigcup \mathcal{D}_2^{\mathrm{N}} \bigcup \ldots \bigcup \mathcal{D}_q^{\mathrm{N}}$ denote the set of all the binary-class training data. Then, we have the following lemma.

**Lemma 2.** *For any $\delta > 0$, the inequalities below hold with probability at least $1 - \delta$:*

$$\sup_{f_1, f_2, \ldots, f_q \in \mathcal{F}} \left| R(f_1, f_2, \ldots, f_q) - \widehat{R}(f_1, f_2, \ldots, f_q) \right| \leq \sum_{k=1}^{q} \left( (1 - \bar{\pi}_k) C_\ell \sqrt{\frac{\ln(2/\delta)}{2n_k^{\mathrm{U}}}} \right.$$

$$\left. + (2 - 2\bar{\pi}_k) L_\ell \mathfrak{R}_{n_k^{\mathrm{U}}, p_k^{\mathrm{U}}}(\mathcal{F}) + (4 - 4\pi_k - 2\bar{\pi}_k) L_\ell \mathfrak{R}_{n_k^{\mathrm{N}}, p_k^{\mathrm{N}}}(\mathcal{F}) + (2 - 2\pi_k - \bar{\pi}_k) C_\ell \sqrt{\frac{\ln(2/\delta)}{2n_k^{\mathrm{N}}}} \right). \tag{17}$$

*Proof.* We can observe that when an unlabeled example $\boldsymbol{x}_{k,i}^{\mathrm{U}} \in \mathcal{D}_k^{\mathrm{U}}$ is substituted by another unlabeled example $\boldsymbol{x}_{k,j}^{\mathrm{U}}$, the value of $\sup_{f_1, f_2, \ldots, f_q \in \mathcal{F}} \left| R(f_1, f_2, \ldots, f_q) - \widehat{R}(f_1, f_2, \ldots, f_q) \right|$ changes at most $(1 - \bar{\pi}_k) C_\ell / n_k^{\mathrm{U}}$. Besides, when a negative example $\boldsymbol{x}_{k,i}^{\mathrm{N}} \in \mathcal{D}_k^{\mathrm{N}}$ is substituted by another negative example $\boldsymbol{x}_{k,j}^{\mathrm{N}}$, the value of $\sup_{f_1, f_2, \ldots, f_q \in \mathcal{F}} \left| R(f_1, f_2, \ldots, f_q) - \widehat{R}(f_1, f_2, \ldots, f_q) \right|$ changes at most $(2 - 2\pi_k - \bar{\pi}_k) C_\ell / n_k^{\mathrm{N}}$. According to the McDiarmid's inequality, for any $\delta > 0$, the following inequality holds with probability at least $1 - \delta/2$:

$$\sup_{f_1, f_2, \ldots, f_q \in \mathcal{F}} \left( R(f_1, f_2, \ldots, f_q) - \widehat{R}(f_1, f_2, \ldots, f_q) \right)$$

$$\leq \mathbb{E}_{\bar{\mathcal{D}}} \left[ \sup_{f_1, f_2, \ldots, f_q \in \mathcal{F}} \left( R(f_1, f_2, \ldots, f_q) - \widehat{R}(f_1, f_2, \ldots, f_q) \right) \right]$$

$$+ \sum_{k=1}^{q} \left( (1 - \bar{\pi}_k) C_\ell \sqrt{\frac{\ln(2/\delta)}{2n_k^{\mathrm{U}}}} + (2 - 2\pi_k - \bar{\pi}_k) C_\ell \sqrt{\frac{\ln(2/\delta)}{2n_k^{\mathrm{N}}}} \right), \tag{18}$$

where the inequality is deduced since $\sqrt{a + b} \leq \sqrt{a} + \sqrt{b}$. It is a routine work to show by symmetrization (Mohri et al., 2012) that

$$\mathbb{E}_{\bar{\mathcal{D}}} \left[ \sup_{f_1, f_2, \ldots, f_q \in \mathcal{F}} \left( R(f_1, f_2, \ldots, f_q) - \widehat{R}(f_1, f_2, \ldots, f_q) \right) \right]$$

$$\leq \sum_{k=1}^{q} \left( (2 - 2\bar{\pi}_k) \mathfrak{R}_{n_k^{\mathrm{U}}, p_k^{\mathrm{U}}}(\ell \circ \mathcal{F}) + (4 - 4\pi_k - 2\bar{\pi}_k) \mathfrak{R}_{n_k^{\mathrm{N}}, p_k^{\mathrm{N}}}(\ell \circ \mathcal{F}) \right), \tag{19}$$

where $\mathfrak{R}_{n,p}(\ell \circ \mathcal{F})$ is the Rademacher complexity of the composite function class $(\ell \circ \mathcal{F})$. According to Talagrand's contraction lemma (Ledoux & Talagrand, 1991), we have

$$\mathfrak{R}_{n_k^{\mathrm{U}}, p_k^{\mathrm{U}}}(\ell \circ \mathcal{F}) \le L_\ell \mathfrak{R}_{n_k^{\mathrm{U}}, p_k^{\mathrm{U}}}(\mathcal{F}), \tag{20}$$

$$\mathfrak{R}_{n_k^{\mathrm{N}}, p_k^{\mathrm{N}}}(\ell \circ \mathcal{F}) \le L_\ell \mathfrak{R}_{n_k^{\mathrm{N}}, p_k^{\mathrm{N}}}(\mathcal{F}). \tag{21}$$

By combining Inequality (18), Inequality (19), Inequality (20), and Inequality (21), the following inequality holds with probability at least $1 - \delta/2$:

$$\sup_{f_1, f_2, \ldots, f_q \in \mathcal{F}} \left( R(f_1, f_2, \ldots, f_q) - \widehat{R}(f_1, f_2, \ldots, f_q) \right) \le \sum_{k=1}^q \left( (2 - 2\bar{\pi}_k) L_\ell \mathfrak{R}_{n_k^{\mathrm{U}}, p_k^{\mathrm{U}}}(\mathcal{F}) \right.$$

$$\left. + (1 - \bar{\pi}_k) C_\ell \sqrt{\frac{\ln(2/\delta)}{2n_k^{\mathrm{U}}}} + (4 - 4\pi_k - 2\bar{\pi}_k) L_\ell \mathfrak{R}_{n_k^{\mathrm{N}}, p_k^{\mathrm{N}}}(\mathcal{F}) + (2 - 2\pi_k - \bar{\pi}_k) C_\ell \sqrt{\frac{\ln(2/\delta)}{2n_k^{\mathrm{N}}}} \right). \tag{22}$$

In the same way, we have the following inequality with probability at least $1 - \delta/2$:

$$\sup_{f_1, f_2, \ldots, f_q \in \mathcal{F}} \left( \widehat{R}(f_1, f_2, \ldots, f_q) - R(f_1, f_2, \ldots, f_q) \right) \le \sum_{k=1}^q \left( (1 - \bar{\pi}_k) C_\ell \sqrt{\frac{\ln(2/\delta)}{2n_k^{\mathrm{U}}}} \right.$$

$$\left. + (2 - 2\bar{\pi}_k) L_\ell \mathfrak{R}_{n_k^{\mathrm{U}}, p_k^{\mathrm{U}}}(\mathcal{F}) + (4 - 4\pi_k - 2\bar{\pi}_k) L_\ell \mathfrak{R}_{n_k^{\mathrm{N}}, p_k^{\mathrm{N}}}(\mathcal{F}) + (2 - 2\pi_k - \bar{\pi}_k) C_\ell \sqrt{\frac{\ln(2/\delta)}{2n_k^{\mathrm{N}}}} \right). \tag{23}$$

By combining Inequality (22) and Inequality (23), we have the following inequality with probability at least $1 - \delta$:

$$\sup_{f_1, f_2, \ldots, f_q \in \mathcal{F}} \left| R(f_1, f_2, \ldots, f_q) - \widehat{R}(f_1, f_2, \ldots, f_q) \right| \le \sum_{k=1}^q \left( (1 - \bar{\pi}_k) C_\ell \sqrt{\frac{\ln(2/\delta)}{2n_k^{\mathrm{U}}}} \right.$$

$$\left. + (2 - 2\bar{\pi}_k) L_\ell \mathfrak{R}_{n_k^{\mathrm{U}}, p_k^{\mathrm{U}}}(\mathcal{F}) + (4 - 4\pi_k - 2\bar{\pi}_k) L_\ell \mathfrak{R}_{n_k^{\mathrm{N}}, p_k^{\mathrm{N}}}(\mathcal{F}) + (2 - 2\pi_k - \bar{\pi}_k) C_\ell \sqrt{\frac{\ln(2/\delta)}{2n_k^{\mathrm{N}}}} \right), \tag{24}$$

which concludes the proof. $\qquad \square$

Finally, the proof of theorem 2 is given.

*Proof of Theorem 2.*

$$R(\widehat{f}_1, \widehat{f}_2, \ldots, \widehat{f}_q) - R(f_1^*, f_2^*, \ldots, f_q^*)$$

$$= R(\widehat{f}_1, \widehat{f}_2, \ldots, \widehat{f}_q) - \widehat{R}(\widehat{f}_1, \widehat{f}_2, \ldots, \widehat{f}_q) + \widehat{R}(\widehat{f}_1, \widehat{f}_2, \ldots, \widehat{f}_q) - \widehat{R}(f_1^*, f_2^*, \ldots, f_q^*)$$

$$\quad + \widehat{R}(f_1^*, f_2^*, \ldots, f_q^*) - R(f_1^*, f_2^*, \ldots, f_q^*)$$

$$\le R(\widehat{f}_1, \widehat{f}_2, \ldots, \widehat{f}_q) - \widehat{R}(\widehat{f}_1, \widehat{f}_2, \ldots, \widehat{f}_q) + \widehat{R}(f_1^*, f_2^*, \ldots, f_q^*) - R(f_1^*, f_2^*, \ldots, f_q^*)$$

$$\le 2 \sup_{f_1, f_2, \ldots, f_q \in \mathcal{F}} \left| R(f_1, f_2, \ldots, f_q) - \widehat{R}(f_1, f_2, \ldots, f_q) \right| \tag{25}$$

The first inequality is deduced because $(\widehat{f}_1, \widehat{f}_2, \ldots, \widehat{f}_q)$ is the minimizer of $\widehat{R}(f_1, f_2, \ldots, f_q)$. Combining Inequality (25) and Lemma 2, the proof is completed. $\qquad \square$

# F  PROOF OF THEOREM 4

Let $\mathfrak{D}_k^+(f_k) = \left\{ (\mathcal{D}_k^{\mathrm{N}}, \mathcal{D}_k^{\mathrm{U}}) \,|\, \widehat{R}_k^{\mathrm{P}}(f_k) \ge 0 \right\}$ and $\mathfrak{D}_k^-(f_k) = \left\{ (\mathcal{D}_k^{\mathrm{N}}, \mathcal{D}_k^{\mathrm{U}}) \,|\, \widehat{R}_k^{\mathrm{P}}(f_k) < 0 \right\}$ denote the sets of NU data pairs having positive and negative empirical risk respectively. Then we have the following lemma.

**Lemma 3.** *The probability measure of $\mathfrak{D}_k^-(f_k)$ can be bounded as follows:*

$$\mathbb{P}\left(\mathfrak{D}_k^-(f_k)\right) \leq \exp\left(\frac{-2\beta^2}{(1-\pi_k-\bar{\pi}_k)^2 C_\ell^2/n_k^{\mathrm{N}} + (1-\bar{\pi}_k)^2 C_\ell^2/n_k^{\mathrm{U}}}\right). \tag{26}$$

*Proof.* Let

$$p\left(\mathcal{D}_k^{\mathrm{N}}\right) = p\left(\boldsymbol{x}_{k,1}^{\mathrm{N}}|\bar{y}_k=1\right) p\left(\boldsymbol{x}_{k,2}^{\mathrm{N}}|\bar{y}_k=1\right)\ldots p\left(\boldsymbol{x}_{k,n_k^{\mathrm{N}}}^{\mathrm{N}}|\bar{y}_k=1\right)$$

and

$$p\left(\mathcal{D}_k^{\mathrm{U}}\right) = p\left(\boldsymbol{x}_{k,1}^{\mathrm{U}}|\bar{y}_k=0\right) p\left(\boldsymbol{x}_{k,2}^{\mathrm{U}}|\bar{y}_k=0\right)\ldots p\left(\boldsymbol{x}_{k,n_k^{\mathrm{U}}}^{\mathrm{U}}|\bar{y}_k=0\right)$$

denote the probability density of $\mathcal{D}_k^{\mathrm{N}}$ and $\mathcal{D}_k^{\mathrm{U}}$ respectively. The joint probability density of $\mathcal{D}_k^{\mathrm{N}}$ and $\mathcal{D}_k^{\mathrm{U}}$ is

$$p\left(\mathcal{D}_k^{\mathrm{N}}, \mathcal{D}_k^{\mathrm{U}}\right) = p\left(\mathcal{D}_k^{\mathrm{N}}\right) p\left(\mathcal{D}_k^{\mathrm{U}}\right).$$

Then, the probability measure $\mathbb{P}\left(\mathfrak{D}_k^-(f_k)\right)$ is defined as

$$\begin{aligned}
\mathbb{P}\left(\mathfrak{D}_k^-(f_k)\right) &= \int_{\left(\mathcal{D}_k^{\mathrm{N}}, \mathcal{D}_k^{\mathrm{U}}\right) \in \mathfrak{D}_k^-(f_k)} p\left(\mathcal{D}_k^{\mathrm{N}}, \mathcal{D}_k^{\mathrm{U}}\right) \mathrm{d}\left(\mathcal{D}_k^{\mathrm{N}}, \mathcal{D}_k^{\mathrm{U}}\right) \\
&= \int_{\left(\mathcal{D}_k^{\mathrm{N}}, \mathcal{D}_k^{\mathrm{U}}\right) \in \mathfrak{D}_k^-(f_k)} p\left(\mathcal{D}_k^{\mathrm{N}}, \mathcal{D}_k^{\mathrm{U}}\right) \mathrm{d}\boldsymbol{x}_{k,1}^{\mathrm{N}}\ldots\mathrm{d}\boldsymbol{x}_{k,n_k^{\mathrm{N}}}^{\mathrm{N}} \mathrm{d}\boldsymbol{x}_{k,1}^{\mathrm{U}}\ldots\mathrm{d}\boldsymbol{x}_{k,n_k^{\mathrm{U}}}^{\mathrm{U}}.
\end{aligned}$$

When a negative example in $\mathcal{D}_k^{\mathrm{N}}$ is substituted by another different negative example, the change of the value of $\widehat{R}_k^{\mathrm{P}}(f_k)$ is no more than $(1-\pi_k-\bar{\pi}_k)C_\ell/n_k^{\mathrm{N}}$; when an unlabeled example in $\mathcal{D}_k^{\mathrm{U}}$ is substituted by another different unlabeled example, the change of the value of $\widehat{R}_k^{\mathrm{P}}(f_k)$ is no more than $(1-\bar{\pi}_k)C_\ell/n_k^{\mathrm{U}}$. Therefore, by applying the McDiarmid's inequality, we can obtain the following inequality:

$$\mathbb{P}\left(\mathbb{E}\left[\widehat{R}_k^{\mathrm{P}}(f_k)\right] - \widehat{R}_k^{\mathrm{P}}(f_k) \geq \beta\right) \leq \exp\left(\frac{-2\beta^2}{(1-\pi_k-\bar{\pi}_k)^2 C_\ell^2/n_k^{\mathrm{N}} + (1-\bar{\pi}_k)^2 C_\ell^2/n_k^{\mathrm{U}}}\right). \tag{27}$$

Therefore, we have

$$\begin{aligned}
\mathbb{P}\left(\mathfrak{D}_k^-(f_k)\right) =& \mathbb{P}\left(\widehat{R}_k^{\mathrm{P}}(f_k) \leq 0\right) \\
\leq& \mathbb{P}\left(\widehat{R}_k^{\mathrm{P}}(f_k) \leq \mathbb{E}\left[\widehat{R}_k^{\mathrm{P}}(f_k)\right] - \beta\right) \\
=& \mathbb{P}\left(\mathbb{E}\left[\widehat{R}_k^{\mathrm{P}}(f_k)\right] - \widehat{R}_k^{\mathrm{P}}(f_k) \geq \beta\right) \\
\leq& \exp\left(\frac{-2\beta^2}{(1-\pi_k-\bar{\pi}_k)^2 C_\ell^2/n_k^{\mathrm{N}} + (1-\bar{\pi}_k)^2 C_\ell^2/n_k^{\mathrm{U}}}\right),
\end{aligned} \tag{28}$$

which concludes the proof. $\qquad\square$

Based on it, the proof of Theorem 3 is provided.

*Proof of Theorem 3.* First, we have

$$\mathbb{E}\left[\widetilde{R}(f_1, f_2, \ldots, f_q)\right] - R(f_1, f_2, \ldots, f_q) = \mathbb{E}\left[\widetilde{R}(f_1, f_2, \ldots, f_q) - \widehat{R}(f_1, f_2, \ldots, f_q)\right].$$

Since $\widetilde{R}(f_1, f_2, \ldots, f_q)$ is an upper bound of $\widehat{R}(f_1, f_2, \ldots, f_q)$, we have

$$\mathbb{E}\left[\widetilde{R}(f_1, f_2, \ldots, f_q)\right] - R(f_1, f_2, \ldots, f_q) \geq 0.$$

Besides, we have

$$
\mathbb{E}\left[\widetilde{R}(f_1, f_2, \ldots, f_q)\right] - R(f_1, f_2, \ldots, f_q)
$$

$$
= \sum_{k=1}^{q} \int_{\left(\mathcal{D}_k^{\mathrm{N}}, \mathcal{D}_k^{\mathrm{U}}\right) \in \mathfrak{D}_k^-(f_k)} \left(g\left(\widehat{R}_k^{\mathrm{P}}(f_k)\right) - \widehat{R}_k^{\mathrm{P}}(f_k)\right) p\left(\mathcal{D}_k^{\mathrm{N}}, \mathcal{D}_k^{\mathrm{U}}\right) \mathrm{d}\left(\mathcal{D}_k^{\mathrm{N}}, \mathcal{D}_k^{\mathrm{U}}\right)
$$

$$
\leq \sum_{k=1}^{q} \sup_{\left(\mathcal{D}_k^{\mathrm{N}}, \mathcal{D}_k^{\mathrm{U}}\right) \in \mathfrak{D}_k^-(f_k)} \left(g\left(\widehat{R}_k^{\mathrm{P}}(f_k)\right) - \widehat{R}_k^{\mathrm{P}}(f_k)\right) \int_{\left(\mathcal{D}_k^{\mathrm{N}}, \mathcal{D}^{\mathrm{U}}\right) \in \mathfrak{D}_k^-(f_k)} p\left(\mathcal{D}_k^{\mathrm{N}}, \mathcal{D}_k^{\mathrm{U}}\right) \mathrm{d}\left(\mathcal{D}_k^{\mathrm{N}}, \mathcal{D}_k^{\mathrm{U}}\right)
$$

$$
= \sum_{k=1}^{q} \sup_{\left(\mathcal{D}_k^{\mathrm{N}}, \mathcal{D}_k^{\mathrm{U}}\right) \in \mathfrak{D}_k^-(f_k)} \left(g\left(\widehat{R}_k^{\mathrm{P}}(f_k)\right) - \widehat{R}_k^{\mathrm{P}}(f_k)\right) \mathbb{P}\left(\mathfrak{D}_k^-(f_k)\right)
$$

$$
\leq \sum_{k=1}^{q} \sup_{\left(\mathcal{D}_k^{\mathrm{N}}, \mathcal{D}_k^{\mathrm{U}}\right) \in \mathfrak{D}_k^-(f_k)} \left(L_g \left|\widehat{R}_k^{\mathrm{P}}(f_k)\right| + \left|\widehat{R}_k^{\mathrm{P}}(f_k)\right|\right) \mathbb{P}\left(\mathfrak{D}_k^-(f_k)\right).
$$

Besides,

$$
\left|\widehat{R}_k^{\mathrm{P}}(f_k)\right| = \left| \frac{\bar{\pi}_k + \pi_k - 1}{n_k^{\mathrm{N}}} \sum_{i=1}^{n_k^{\mathrm{N}}} \ell\left(f_k(\boldsymbol{x}_{k,i}^{\mathrm{N}})\right) + \frac{1 - \bar{\pi}_k}{n_k^{\mathrm{U}}} \sum_{i=1}^{n_k^{\mathrm{U}}} \ell\left(f_k(\boldsymbol{x}_{k,i}^{\mathrm{U}})\right) \right|
$$

$$
\leq \left| \frac{\bar{\pi}_k + \pi_k - 1}{n_k^{\mathrm{N}}} \sum_{i=1}^{n_k^{\mathrm{N}}} \ell\left(f_k(\boldsymbol{x}_{k,i}^{\mathrm{N}})\right) \right| + \left| \frac{1 - \bar{\pi}_k}{n_k^{\mathrm{U}}} \sum_{i=1}^{n_k^{\mathrm{U}}} \ell\left(f_k(\boldsymbol{x}_{k,i}^{\mathrm{U}})\right) \right|
$$

$$
\leq (1 - \pi_k - \bar{\pi}_k) C_\ell + (1 - \bar{\pi}_k) C_\ell = (2 - 2\bar{\pi}_k - \pi_k) C_\ell.
$$

Therefore, we have

$$
\mathbb{E}\left[\widetilde{R}(f_1, f_2, \ldots, f_q)\right] - R(f_1, f_2, \ldots, f_q)
$$

$$
\leq \sum_{k=1}^{q} \sup_{\left(\mathcal{D}_k^{\mathrm{N}}, \mathcal{D}_k^{\mathrm{U}}\right) \in \mathfrak{D}_k^-(f_k)} \left(L_g \left|\widehat{R}_k^{\mathrm{P}}(f_k)\right| + \left|\widehat{R}_k^{\mathrm{P}}(f_k)\right|\right) \mathbb{P}(\mathfrak{D}_k^-(f_k)).
$$

$$
\leq \sum_{k=1}^{q} (2 - 2\bar{\pi}_k - \pi_k)(L_g + 1) C_\ell \exp\left(\frac{-2\beta^2}{(1 - \pi_k - \bar{\pi}_k)^2 C_\ell^2 / n_k^{\mathrm{N}} + (1 - \bar{\pi}_k)^2 C_\ell^2 / n_k^{\mathrm{U}}}\right)
$$

$$
= \sum_{k=1}^{q} (2 - 2\bar{\pi}_k - \pi_k)(L_g + 1) C_\ell \Delta_k,
$$

which concludes the first part of the proof of Theorem 3. Before giving the proof for the second part, we give the upper bound of $\left|\mathbb{E}\left[\widetilde{R}(f_1, f_2, \ldots, f_q)\right] - \widetilde{R}(f_1, f_2, \ldots, f_q)\right|$. When an unlabeled example from $\mathcal{D}_k^{\mathrm{U}}$ is substituted by another unlabeled example, the value of $\widetilde{R}(f_1, f_2, \ldots, f_q)$ changes at most $(1 - \bar{\pi}_k) C_\ell L_g / n_k^{\mathrm{U}}$. When a negative example from $\mathcal{D}_k^{\mathrm{N}}$ is substituted by a different example, the value of $\widetilde{R}(f_1, f_2, \ldots, f_q)$ changes at most $((1 - \pi_k - \bar{\pi}_k) L_g + 1 - \pi_k) C_\ell / n_k^{\mathrm{N}}$. By applying McDiarmid's inequality, we have the following inequalities with probability at least $1 - \delta/2$:

$$
\widetilde{R}(f_1, f_2, \ldots, f_q) - \mathbb{E}\left[\widetilde{R}(f_1, f_2, \ldots, f_q)\right] \leq \sum_{k=1}^{q} (1 - \bar{\pi}_k) C_\ell L_g \sqrt{\frac{\ln(2/\delta)}{2n_k^{\mathrm{U}}}}
$$

$$
+ \sum_{k=1}^{q} ((1 - \pi_k - \bar{\pi}_k) L_g + 1 - \pi_k) C_\ell \sqrt{\frac{\ln(2/\delta)}{2n_k^{\mathrm{N}}}},
$$

$$
\mathbb{E}\left[\widetilde{R}(f_1, f_2, \ldots, f_q)\right] - \widetilde{R}(f_1, f_2, \ldots, f_q) \leq \sum_{k=1}^{q} (1 - \bar{\pi}_k) C_\ell L_g \sqrt{\frac{\ln(2/\delta)}{2n_k^{\mathrm{U}}}}
$$

$$
+ \sum_{k=1}^{q} ((1 - \pi_k - \bar{\pi}_k) L_g + 1 - \pi_k) C_\ell \sqrt{\frac{\ln(2/\delta)}{2n_k^{\mathrm{N}}}}.
$$

Then, with probability at least $1 - \delta$, we have

$$\left| \mathbb{E}\left[ \widetilde{R}(f_1, f_2, \ldots, f_q) \right] - \widetilde{R}(f_1, f_2, \ldots, f_q) \right| \leq \sum_{k=1}^{q} (1 - \bar{\pi}_k) C_\ell L_g \sqrt{\frac{\ln(2/\delta)}{2n_k^{\mathrm{U}}}}$$
$$+ \sum_{k=1}^{q} ((1 - \pi_k - \bar{\pi}_k) L_g + 1 - \pi_k) C_\ell \sqrt{\frac{\ln(2/\delta)}{2n_k^{\mathrm{N}}}}.$$

Therefore, with probability at least $1 - \delta$ we have

$$\left| \widetilde{R}(f_1, f_2, \ldots, f_q) - R(f_1, f_2, \ldots, f_q) \right|$$
$$= \left| \widetilde{R}(f_1, f_2, \ldots, f_q) - \mathbb{E}[\widetilde{R}(f_1, f_2, \ldots, f_q)] + \mathbb{E}[\widetilde{R}(f_1, f_2, \ldots, f_q)] - R(f_1, f_2, \ldots, f_q) \right|$$
$$\leq \left| \widetilde{R}(f_1, f_2, \ldots, f_q) - \mathbb{E}[\widetilde{R}(f_1, f_2, \ldots, f_q)] \right| + \left| \mathbb{E}[\widetilde{R}(f_1, f_2, \ldots, f_q)] - R(f_1, f_2, \ldots, f_q) \right|$$
$$= \left| \widetilde{R}(f_1, f_2, \ldots, f_q) - \mathbb{E}[\widetilde{R}(f_1, f_2, \ldots, f_q)] \right| + \mathbb{E}[\widetilde{R}(f_1, f_2, \ldots, f_q)] - R(f_1, f_2, \ldots, f_q)$$
$$\leq \sum_{k=1}^{q} (1 - \bar{\pi}_k) C_\ell L_g \sqrt{\frac{\ln(2/\delta)}{2n_k^{\mathrm{U}}}} + \sum_{k=1}^{q} ((1 - \pi_k - \bar{\pi}_k) L_g + 1 - \pi_k) C_\ell \sqrt{\frac{\ln(2/\delta)}{2n_k^{\mathrm{N}}}}$$
$$+ \sum_{k=1}^{q} (2 - 2\bar{\pi}_k - \pi_k) (L_g + 1) C_\ell \Delta_k,$$

which concludes the proof. $\qquad\square$

## G    PROOF OF THEOREM 5

In this section, we adopt an alternative definition of Rademacher complexity:

$$\mathfrak{R}'_{n,p}(\mathcal{F}) = \mathbb{E}_{\mathcal{X}_n} \mathbb{E}_{\boldsymbol{\sigma}} \left[ \sup_{f \in \mathcal{F}} \left| \frac{1}{n} \sum_{i=1}^{n} \sigma_i f(\boldsymbol{x}_i) \right| \right]. \tag{29}$$

Then, we introduce the following lemmas.

**Lemma 4.** *Without any composition, for any $\mathcal{F}$, we have $\mathfrak{R}'_{n,p}(\mathcal{F}) \geq \mathfrak{R}_{n,p}(\mathcal{F})$. If $\mathcal{F}$ is closed under negation, we have $\mathfrak{R}'_{n,p}(\mathcal{F}) = \mathfrak{R}_{n,p}(\mathcal{F})$.*

**Lemma 5** (Theorem 4.12 in (Ledoux & Talagrand, 1991)). *If $\psi : \mathbb{R} \to \mathbb{R}$ is a Lipschitz continuous function with a Lipschitz constant $L_\psi$ and satisfies $\psi(0) = 0$, we have*

$$\mathfrak{R}'_{n,p}(\psi \circ \mathcal{F}) \leq 2L_\psi \mathfrak{R}'_{n,p}(\mathcal{F}),$$

*where $\psi \circ \mathcal{F} = \{\psi \circ f | f \in \mathcal{F}\}$.*

Before giving the proof of Theorem 5, we give the following lemma.

**Lemma 6.** *For any $\delta > 0$, the inequalities below hold with probability at least $1 - \delta$:*

$$\sup_{f_1, f_2, \ldots, f_q \in \mathcal{F}} \left| R(f_1, f_2, \ldots, f_q) - \widetilde{R}(f_1, f_2, \ldots, f_q) \right|$$
$$\leq \sum_{k=1}^{q} (1 - \bar{\pi}_k) C_\ell L_g \sqrt{\frac{\ln(1/\delta)}{2n_k^{\mathrm{U}}}} + \sum_{k=1}^{q} ((1 - \pi_k - \bar{\pi}_k) L_g + 1 - \pi_k) C_\ell \sqrt{\frac{\ln(1/\delta)}{2n_k^{\mathrm{N}}}}$$
$$+ \sum_{k=1}^{q} \left( (4 - 4\bar{\pi}_k) L_g L_\ell \mathfrak{R}_{n_k^{\mathrm{U}}, p_k^{\mathrm{U}}}(\mathcal{F}) + ((4 - 4\pi_k - 4\bar{\pi}_k) L_g + 4 - 4\pi_k) L_\ell \mathfrak{R}_{n_k^{\mathrm{N}}, p_k^{\mathrm{N}}}(\mathcal{F}) \right)$$
$$+ \sum_{k=1}^{q} (2 - 2\bar{\pi}_k - \pi_k) (L_g + 1) C_\ell \Delta_k.$$

*Proof.* Similar to previous proofs, we can observe that when an unlabeled example from $\mathcal{D}_k^{\mathrm{U}}$ is substituted by another unlabeled example, the value of $\sup_{f_1, f_2, \ldots, f_q \in \mathcal{F}} \left| \mathbb{E}\left[ \widetilde{R}(f_1, f_2, \ldots, f_q) \right] - \widetilde{R}(f_1, f_2, \ldots, f_q) \right|$ changes at most $(1 - \bar{\pi}_k) C_\ell L_g / n_k^{\mathrm{U}}$. When a negative example from $\mathcal{D}_k^{\mathrm{N}}$ is substituted by a different example, the value of $\sup_{f_1, f_2, \ldots, f_q \in \mathcal{F}} \left| \mathbb{E}\left[ \widetilde{R}(f_1, f_2, \ldots, f_q) \right] - \widetilde{R}(f_1, f_2, \ldots, f_q) \right|$ changes at most $((1 - \pi_k - \bar{\pi}_k) L_g + 1 - \pi_k) C_\ell / n_k^{\mathrm{N}}$. By applying McDiarmid's inequality, we have the following inequality with probability at least $1 - \delta$:

$$
\begin{aligned}
&\sup_{f_1, f_2, \ldots, f_q \in \mathcal{F}} \left| \mathbb{E}\left[ \widetilde{R}(f_1, f_2, \ldots, f_q) \right] - \widetilde{R}(f_1, f_2, \ldots, f_q) \right| \\
&- \mathbb{E}\left[ \sup_{f_1, f_2, \ldots, f_q \in \mathcal{F}} \left| \mathbb{E}\left[ \widetilde{R}(f_1, f_2, \ldots, f_q) \right] - \widetilde{R}(f_1, f_2, \ldots, f_q) \right| \right] \\
&\leq \sum_{k=1}^q (1 - \bar{\pi}_k) C_\ell L_g \sqrt{\frac{\ln(1/\delta)}{2 n_k^{\mathrm{U}}}} + \sum_{k=1}^q ((1 - \pi_k - \bar{\pi}_k) L_g + 1 - \pi_k) C_\ell \sqrt{\frac{\ln(1/\delta)}{2 n_k^{\mathrm{N}}}}.
\end{aligned} \tag{30}
$$

Besides, we have

$$
\begin{aligned}
&\mathbb{E}\left[ \sup_{f_1, f_2, \ldots, f_q \in \mathcal{F}} \left| \mathbb{E}\left[ \widetilde{R}(f_1, f_2, \ldots, f_q) \right] - \widetilde{R}(f_1, f_2, \ldots, f_q) \right| \right] \\
&= \mathbb{E}_{\bar{\mathcal{D}}}\left[ \sup_{f_1, f_2, \ldots, f_q \in \mathcal{F}} \left| \mathbb{E}_{\bar{\mathcal{D}}'}\left[ \widetilde{R}(f_1, f_2, \ldots, f_q) \right] - \widetilde{R}(f_1, f_2, \ldots, f_q) \right| \right] \\
&\leq \mathbb{E}_{\bar{\mathcal{D}}, \bar{\mathcal{D}}'}\left[ \sup_{f_1, f_2, \ldots, f_q \in \mathcal{F}} \left| \widetilde{R}(f_1, f_2, \ldots, f_q; \bar{\mathcal{D}}) - \widetilde{R}(f_1, f_2, \ldots, f_q; \bar{\mathcal{D}}') \right| \right],
\end{aligned} \tag{31}
$$

where the last inequality is deduced by applying Jensen's inequality twice since the absolute value function and the supremum function are both convex. Here, $\widetilde{R}(f_1, f_2, \ldots, f_q; \bar{\mathcal{D}})$ denotes the value of $\widetilde{R}(f_1, f_2, \ldots, f_q)$ calculated on $\bar{\mathcal{D}}$. To ensure that the conditions in Lemma 5 hold, we introduce $\tilde{\ell}(z) = \ell(z) - \ell(0)$. It is obvious that $\tilde{\ell}(0) = 0$ and $\tilde{\ell}(z)$ is also a Lipschitz continuous function with a Lipschitz constant $L_\ell$. Then, we have

$$
\begin{aligned}
&\left| \widetilde{R}(f_1, f_2, \ldots, f_q; \bar{\mathcal{D}}) - \widetilde{R}(f_1, f_2, \ldots, f_q; \bar{\mathcal{D}}') \right| \\
&\leq \sum_{k=1}^q \left| g\left( \frac{\bar{\pi}_k + \pi_k - 1}{n_k^{\mathrm{N}}} \sum_{i=1}^{n_k^{\mathrm{N}}} \ell\left( f_k(\boldsymbol{x}_{k,i}^{\mathrm{N}}) \right) + \frac{1 - \bar{\pi}_k}{n_k^{\mathrm{U}}} \sum_{i=1}^{n_k^{\mathrm{U}}} \ell\left( f_k(\boldsymbol{x}_{k,i}^{\mathrm{U}}) \right) \right) \right. \\
&\qquad \left. - g\left( \frac{\bar{\pi}_k + \pi_k - 1}{n_k^{\mathrm{N}}} \sum_{i=1}^{n_k^{\mathrm{N}}} \ell\left( f_k(\boldsymbol{x}_{k,i}^{\mathrm{N}'}) \right) + \frac{1 - \bar{\pi}_k}{n_k^{\mathrm{U}}} \sum_{i=1}^{n_k^{\mathrm{U}}} \ell\left( f_k(\boldsymbol{x}_{k,i}^{\mathrm{U}'}) \right) \right) \right| \\
&\quad + \sum_{k=1}^q \left| \frac{1 - \pi_k}{n_k^{\mathrm{N}}} \sum_{i=1}^{n_k^{\mathrm{N}}} \ell\left( -f_k(\boldsymbol{x}_{k,i}^{\mathrm{N}}) \right) - \frac{1 - \pi_k}{n_k^{\mathrm{N}}} \sum_{i=1}^{n_k^{\mathrm{N}}} \ell\left( -f_k(\boldsymbol{x}_{k,i}^{\mathrm{N}'}) \right) \right| \\
&\leq \sum_{k=1}^q L_g \left| \frac{\bar{\pi}_k + \pi_k - 1}{n_k^{\mathrm{N}}} \sum_{i=1}^{n_k^{\mathrm{N}}} \left( \ell\left( f_k(\boldsymbol{x}_{k,i}^{\mathrm{N}}) \right) - \ell\left( f_k(\boldsymbol{x}_{k,i}^{\mathrm{N}'}) \right) \right) + \frac{1 - \bar{\pi}_k}{n_k^{\mathrm{N}}} \sum_{i=1}^{n_k^{\mathrm{U}}} \left( \ell\left( f_k(\boldsymbol{x}_{k,i}^{\mathrm{U}}) \right) - \ell\left( f_k(\boldsymbol{x}_{k,i}^{\mathrm{U}'}) \right) \right) \right| \\
&\quad + \sum_{k=1}^q \left| \frac{1 - \pi_k}{n_k^{\mathrm{N}}} \sum_{i=1}^{n_k^{\mathrm{N}}} \left( \ell\left( -f_k(\boldsymbol{x}_{k,i}^{\mathrm{N}}) \right) - \ell\left( -f_k(\boldsymbol{x}_{k,i}^{\mathrm{N}'}) \right) \right) \right|.
\end{aligned} \tag{32}
$$

Besides, we can observe $\ell(z_1) - \ell(z_2) = \tilde{\ell}(z_1) - \tilde{\ell}(z_2)$. Therefore, the RHS of Inequality (32) can be expressed as

$$
\sum_{k=1}^{q} L_g \left| \frac{\bar{\pi}_k + \pi_k - 1}{n_k^{\mathrm{N}}} \sum_{i=1}^{n_k^{\mathrm{N}}} \left( \tilde{\ell}\left(f_k(\boldsymbol{x}_{k,i}^{\mathrm{N}})\right) - \tilde{\ell}\left(f_k(\boldsymbol{x}_{k,i}^{\mathrm{N}'})\right) \right) + \frac{1 - \bar{\pi}_k}{n_k^{\mathrm{N}}} \sum_{i=1}^{n_k^{\mathrm{U}}} \left( \tilde{\ell}\left(f_k(\boldsymbol{x}_{k,i}^{\mathrm{U}})\right) - \tilde{\ell}\left(f_k(\boldsymbol{x}_{k,i}^{\mathrm{U}'})\right) \right) \right|
$$

$$
+ \sum_{k=1}^{q} \left| \frac{1 - \pi_k}{n_k^{\mathrm{N}}} \sum_{i=1}^{n_k^{\mathrm{N}}} \left( \tilde{\ell}\left(-f_k(\boldsymbol{x}_{k,i}^{\mathrm{N}})\right) - \tilde{\ell}\left(-f_k(\boldsymbol{x}_{k,i}^{\mathrm{N}'})\right) \right) \right|.
$$

Then, it is a routine work to show by symmetrization (Mohri et al., 2012) that

$$
\mathbb{E}_{\bar{\mathcal{D}},\mathcal{D}'} \left[ \sup_{f_1,f_2,\ldots,f_q \in \mathcal{F}} \left| \widetilde{R}(f_1,f_2,\ldots,f_q; \bar{\mathcal{D}}) - \widetilde{R}(f_1,f_2,\ldots,f_q; \mathcal{D}') \right| \right]
$$

$$
\leq \sum_{k=1}^{q} \left( (2 - 2\bar{\pi}_k) L_g \mathfrak{R}'_{n_k^{\mathrm{U}},p_k^{\mathrm{U}}}(\tilde{\ell} \circ \mathcal{F}) + ((2 - 2\pi_k - 2\bar{\pi}_k) L_g + 2 - 2\pi_k) \mathfrak{R}'_{n_k^{\mathrm{N}},p_k^{\mathrm{N}}}(\tilde{\ell} \circ \mathcal{F}) \right)
$$

$$
\leq \sum_{k=1}^{q} \left( (4 - 4\bar{\pi}_k) L_g L_\ell \mathfrak{R}'_{n_k^{\mathrm{U}},p_k^{\mathrm{U}}}(\mathcal{F}) + ((4 - 4\pi_k - 4\bar{\pi}_k) L_g + 4 - 4\pi_k) L_\ell \mathfrak{R}'_{n_k^{\mathrm{N}},p_k^{\mathrm{N}}}(\mathcal{F}) \right)
$$

$$
= \sum_{k=1}^{q} \left( (4 - 4\bar{\pi}_k) L_g L_\ell \mathfrak{R}_{n_k^{\mathrm{U}},p_k^{\mathrm{U}}}(\mathcal{F}) + ((4 - 4\pi_k - 4\bar{\pi}_k) L_g + 4 - 4\pi_k) L_\ell \mathfrak{R}_{n_k^{\mathrm{N}},p_k^{\mathrm{N}}}(\mathcal{F}) \right), \quad (33)
$$

where the second inequality is deduced according to Lemma 5 and the last equality is based on the assumption that $\mathcal{F}$ is closed under negation. By combing Inequality (30), Inequality (31), and Inequality (33), we have the following inequality with probability at least $1 - \delta$:

$$
\sup_{f_1,f_2,\ldots,f_q \in \mathcal{F}} \left| \mathbb{E}\left[ \widetilde{R}(f_1,f_2,\ldots,f_q) \right] - \widetilde{R}(f_1,f_2,\ldots,f_q) \right|
$$

$$
\leq \sum_{k=1}^{q} (1 - \bar{\pi}_k) C_\ell L_g \sqrt{\frac{\ln(1/\delta)}{2n_k^{\mathrm{U}}}} + \sum_{k=1}^{q} ((1 - \pi_k - \bar{\pi}_k) L_g + 1 - \pi_k) C_\ell \sqrt{\frac{\ln(1/\delta)}{2n_k^{\mathrm{N}}}}
$$

$$
+ \sum_{k=1}^{q} \left( (4 - 4\bar{\pi}_k) L_g L_\ell \mathfrak{R}_{n_k^{\mathrm{U}},p_k^{\mathrm{U}}}(\mathcal{F}) + ((4 - 4\pi_k - 4\bar{\pi}_k) L_g + 4 - 4\pi_k) L_\ell \mathfrak{R}_{n_k^{\mathrm{N}},p_k^{\mathrm{N}}}(\mathcal{F}) \right). \quad (34)
$$

Then, we have

$$
\sup_{f_1,f_2,\ldots,f_q \in \mathcal{F}} \left| R(f_1,f_2,\ldots,f_q) - \widetilde{R}(f_1,f_2,\ldots,f_q) \right|
$$

$$
= \sup_{f_1,f_2,\ldots,f_q \in \mathcal{F}} \left| R(f_1,f_2,\ldots,f_q) - \mathbb{E}\left[ \widetilde{R}(f_1,f_2,\ldots,f_q) \right] \right.
$$

$$
\left. + \mathbb{E}\left[ \widetilde{R}(f_1,f_2,\ldots,f_q) \right] - \widetilde{R}(f_1,f_2,\ldots,f_q) \right|
$$

$$
\leq \sup_{f_1,f_2,\ldots,f_q \in \mathcal{F}} \left| R(f_1,f_2,\ldots,f_q) - \mathbb{E}\left[ \widetilde{R}(f_1,f_2,\ldots,f_q) \right] \right|
$$

$$
+ \sup_{f_1,f_2,\ldots,f_q \in \mathcal{F}} \left| \mathbb{E}\left[ \widetilde{R}(f_1,f_2,\ldots,f_q) \right] - \widetilde{R}(f_1,f_2,\ldots,f_q) \right|. \quad (35)
$$

Combining Inequality (35) with Inequality (34) and Inequality (10), the proof is completed. $\qquad\square$

Then, we give the proof of Theorem 5.

*Proof of Theorem 5.*

$$
\begin{aligned}
& R(\widetilde{f}_1, \widetilde{f}_2, \ldots, \widetilde{f}_q) - R(f_1^*, f_2^*, \ldots, f_q^*) \\
=& R(\widetilde{f}_1, \widetilde{f}_2, \ldots, \widetilde{f}_q) - \widetilde{R}(\widetilde{f}_1, \widetilde{f}_2, \ldots, \widetilde{f}_q) + \widetilde{R}(\widetilde{f}_1, \widetilde{f}_2, \ldots, \widetilde{f}_q) - \widetilde{R}(f_1^*, f_2^*, \ldots, f_q^*) \\
& + \widetilde{R}(f_1^*, f_2^*, \ldots, f_q^*) - R(f_1^*, f_2^*, \ldots, f_q^*) \\
\leq& R(\widetilde{f}_1, \widetilde{f}_2, \ldots, \widetilde{f}_q) - \widetilde{R}(\widetilde{f}_1, \widetilde{f}_2, \ldots, \widetilde{f}_q) + \widetilde{R}(f_1^*, f_2^*, \ldots, f_q^*) - R(f_1^*, f_2^*, \ldots, f_q^*) \\
\leq& 2 \sup_{f_1, f_2, \ldots, f_q \in \mathcal{F}} \left| R(f_1, f_2, \ldots, f_q) - \widetilde{R}(f_1, f_2, \ldots, f_q) \right|.
\end{aligned}
\tag{36}
$$

The first inequality is deduced because $(\widetilde{f}_1, \widetilde{f}_2, \ldots, \widetilde{f}_q)$ is the minimizer of $\widetilde{R}(f_1, f_2, \ldots, f_q)$. Combining Inequality (36) and Lemma 6, the proof is completed. $\square$

# H DETAILS OF EXPERIMENTAL SETUP

## H.1 DETAILS OF SYNTHETIC BENCHMARK DATASETS

For the "uniform" setting, a label other than the ground-truth label was sampled randomly following the uniform distribution to be the complementary label.

For the "biased-a" and "biased-b" settings, we adopted the following row-normalized transition matrices of $p(\bar{y}|y)$ to generate complementary labels:

$$
\text{biased-a:} \begin{bmatrix}
0 & 0.250 & 0.043 & 0.040 & 0.043 & 0.040 & 0.250 & 0.040 & 0.250 & 0.043 \\
0.043 & 0 & 0.250 & 0.043 & 0.040 & 0.043 & 0.040 & 0.250 & 0.040 & 0.250 \\
0.250 & 0.043 & 0 & 0.250 & 0.043 & 0.040 & 0.043 & 0.040 & 0.250 & 0.040 \\
0.040 & 0.250 & 0.043 & 0 & 0.250 & 0.043 & 0.040 & 0.043 & 0.040 & 0.250 \\
0.250 & 0.040 & 0.250 & 0.043 & 0 & 0.250 & 0.043 & 0.040 & 0.043 & 0.040 \\
0.040 & 0.250 & 0.040 & 0.250 & 0.043 & 0 & 0.250 & 0.043 & 0.040 & 0.043 \\
0.043 & 0.040 & 0.250 & 0.040 & 0.250 & 0.043 & 0 & 0.250 & 0.043 & 0.040 \\
0.040 & 0.043 & 0.040 & 0.250 & 0.040 & 0.250 & 0.043 & 0 & 0.250 & 0.043 \\
0.043 & 0.040 & 0.043 & 0.040 & 0.250 & 0.040 & 0.250 & 0.043 & 0 & 0.250 \\
0.250 & 0.043 & 0.040 & 0.043 & 0.040 & 0.250 & 0.040 & 0.250 & 0.043 & 0
\end{bmatrix},
$$

$$
\text{biased-b:} \begin{bmatrix}
0 & 0.220 & 0.080 & 0.033 & 0.080 & 0.033 & 0.220 & 0.033 & 0.220 & 0.080 \\
0.080 & 0 & 0.220 & 0.080 & 0.033 & 0.080 & 0.033 & 0.220 & 0.033 & 0.220 \\
0.220 & 0.080 & 0 & 0.220 & 0.080 & 0.033 & 0.080 & 0.033 & 0.220 & 0.033 \\
0.033 & 0.220 & 0.080 & 0 & 0.220 & 0.080 & 0.033 & 0.080 & 0.033 & 0.220 \\
0.220 & 0.033 & 0.220 & 0.080 & 0 & 0.220 & 0.080 & 0.033 & 0.080 & 0.033 \\
0.033 & 0.220 & 0.033 & 0.220 & 0.080 & 0 & 0.220 & 0.080 & 0.033 & 0.080 \\
0.080 & 0.033 & 0.220 & 0.033 & 0.220 & 0.080 & 0 & 0.220 & 0.080 & 0.033 \\
0.033 & 0.080 & 0.033 & 0.220 & 0.033 & 0.220 & 0.080 & 0 & 0.220 & 0.080 \\
0.080 & 0.033 & 0.080 & 0.033 & 0.220 & 0.033 & 0.220 & 0.080 & 0 & 0.220 \\
0.220 & 0.080 & 0.033 & 0.080 & 0.033 & 0.220 & 0.033 & 0.220 & 0.080 & 0
\end{bmatrix}.
$$

For each example, we sample a complementary label from a multinomial distribution parameterized by the row vector of the transition matrix indexed by the ground-truth label.

For the "SCAR-a" and "SCAR-b" settings, we followed the generation process in Section 3.1 with the following class priors of complementary labels:

$$
\text{SCAR-a: } [0.05, 0.05, 0.2, 0.2, 0.1, 0.1, 0.05, 0.05, 0.1, 0.1],
$$
$$
\text{SCAR-b: } [0.1, 0.1, 0.2, 0.05, 0.05, 0.1, 0.1, 0.2, 0.05, 0.05].
$$

We repeated the sampling procedure to ensure that each example had a single complementary label.

## H.2 DESCRIPTIONS OF COMPARED APPROACHES

The compared methods in the experiments of synthetic benchmark datasets:

- PC (Ishida et al., 2017): A risk-consistent complementary-label learning approach using the pairwise comparison loss.
- NN (Ishida et al., 2019): A risk-consistent complementary-label learning approach using the non-negative risk estimator.
- GA (Ishida et al., 2019): A variant of the non-negative risk estimator of complementary-label learning by using the gradient ascent technique.
- L-UW (Gao & Zhang, 2021): A discriminative approach by minimizing the outputs corresponding to complementary labels.
- L-W Gao & Zhang (2021): A weighted loss based on L-UW by considering the prediction uncertainty.
- OP (Liu et al., 2023): A classifier-consistent complementary-label learning approach by minimizing the outputs of complementary labels.

The compared methods in the experiments of real-world benchmark datasets:

- CC (Feng et al., 2020b): A classifier-consistent partial-label learning approach based on the uniform distribution assumption of partial labels.
- PRODEN (Lv et al., 2020): A risk-consistent partial-label learning approach using the self-training strategy to identify the ground-truth labels.
- EXP (Feng et al., 2020a): A classifier-consistent multiple complementary-label learning approach by using the exponential loss function.
- MAE (Feng et al., 2020a): A classifier-consistent multiple complementary-label learning approach by using the Mean Absolute Error loss function.
- Phuber-CE (Feng et al., 2020a): A classifier-consistent multiple complementary-label learning approach by using the Partially Huberised Cross Entropy loss function.
- LWS (Wen et al., 2021): A partial-label learning approach by leveraging a weight to account for the tradeoff between losses on partial and non-partial labels.
- CAVL (Zhang et al., 2022): A partial-label learning approach by using the class activation value to identify the true labels.
- IDGP (Qiao et al., 2023): A instance-dependent partial-label learning approach by modeling the generation process of partial labels.

### H.3 Details of Models and Hyperparameters

The logistic loss was adopted to instantiate the binary-class loss function $l$, and the absolute value function was used as the risk correction function $g$ for CONU.

For CIFAR-10, we used 34-layer ResNet (He et al., 2016) and 22-layer DenseNet (Huang et al., 2017) as the model architectures. For the other three datasets, we used a multilayer perceptron (MLP) with three hidden layers of width 300 equipped with the ReLU (Nair & Hinton, 2010) activation function and batch normalization (Ioffe & Szegedy, 2015) and 5-layer LeNet (LeCun et al., 1998) as the model architectures.

For CLCIFAR-10 and CLCIFAR-20, we adopted the same data augmentation techniques for all the methods, including random horizontal flipping and random cropping. We used 34-layer ResNet (He et al., 2016) and 22-layer DenseNet (Huang et al., 2017) as the model architectures.

All the methods were implemented in PyTorch (Paszke et al., 2019). We used the Adam optimizer (Kingma & Ba, 2015). The learning rate and batch size were fixed to 1e-3 and 256 for all the datasets, respectively. The weight decay was 1e-3 for CIFAR-10, CLCIFAR-10, and CLCIFAR-20 and 1e-5 for the other three datasets. The number of epochs was set to 200, and we recorded the mean accuracy in the last ten epochs.

## I A brief review of PU learning approaches

The goal of PU learning is to learn a binary classifier from positive and unlabeled data only. PU learning methods can be broadly classified into two groups: sample selection methods and cost-sensitive methods. Sample selection methods try to identify negative examples from the unlabeled dataset and then use supervised learning methods to learn the classifier (Wang et al., 2023; Dai et al., 2023; Garg et al., 2021). Cost-sensitive methods are based on the unbiased risk estimator,

which rewrites the classification risk as that only on positive and unlabeled data (Kiryo et al., 2017; Jiang et al., 2023; Zhao et al., 2022).

## J    MORE EXPERIMENTAL RESULTS

### J.1    EXPERIMENTAL RESULTS ON CIFAR-10

Table 6: Classification accuracy (mean±std) of each method on CIFAR-10 with single complementary label. The best performance is shown in bold (pairwise *t*-test at the 0.05 significance level).

| Setting | Uniform | | Biased-a | | Biased-b | | SCAR-a | | SCAR-b | |
|---|---|---|---|---|---|---|---|---|---|---|
| Model | ResNet | DenseNet | ResNet | DenseNet | ResNet | DenseNet | ResNet | DenseNet | ResNet | DenseNet |
| PC | 14.33 ±0.73 | 17.44 ±0.52 | 25.46 ±0.69 | 34.01 ±1.47 | 23.04 ±0.33 | 29.27 ±1.05 | 14.94 ±0.88 | 17.11 ±0.87 | 17.16 ±0.86 | 21.14 ±1.34 |
| NN | 19.90 ±0.73 | 30.55 ±1.01 | 24.88 ±1.01 | 24.48 ±1.50 | 26.59 ±1.33 | 24.51 ±1.24 | 21.11 ±0.94 | 29.48 ±1.05 | 23.56 ±1.25 | 30.67 ±0.73 |
| GA | **37.59** **±1.76** | **46.86** **±0.84** | 20.01 ±1.96 | 22.41 ±1.33 | 16.74 ±2.64 | 21.48 ±1.46 | **24.17** **±1.32** | 29.04 ±1.84 | 23.47 ±1.30 | 30.72 ±1.44 |
| L-UW | 19.58 ±1.77 | 17.25 ±3.03 | 24.83 ±2.67 | 29.46 ±1.03 | 20.73 ±2.41 | 25.41 ±2.61 | 14.56 ±2.71 | 10.69 ±0.94 | 10.39 ±0.50 | 10.04 ±0.09 |
| L-W | 18.05 ±3.02 | 13.97 ±2.55 | 22.65 ±2.70 | 27.64 ±0.80 | 22.70 ±2.33 | 24.86 ±1.34 | 13.72 ±2.60 | 10.00 ±0.00 | 10.25 ±0.49 | 10.00 ±0.00 |
| OP | 23.78 ±2.80 | 39.32 ±2.46 | 29.47 ±2.71 | 41.99 ±1.54 | 25.60 ±4.18 | 39.61 ±2.26 | 17.55 ±1.38 | 27.12 ±1.17 | 20.08 ±2.96 | 27.24 ±2.62 |
| CONU | **35.63** **±3.23** | 42.65 ±2.00 | **39.70** **±3.79** | **51.42** **±1.81** | **37.82** **±2.72** | **50.52** **±2.18** | **29.04** **±3.70** | **36.38** **±2.56** | **35.71** **±1.16** | **38.43** **±0.85** |

### J.2    ANALYSIS OF EXPERIMENTAL RESULTS

On synthetic benchmark datasets, we can observe

- Out of 40 cases of different distributions and datasets, CONU achieves the best performance in 39 cases, which clearly validates the effectiveness of the proposed approach.
- Some consistent approaches based on the uniform distribution assumption can achieve comparable or better or comparable performance to CONU for the "uniform" setting. For example, GA outperforms CONU on CIFAR-10. However, its performance drops significantly on other distribution settings. This shows that methods based on the uniform distribution assumption cannot handle more realistic data distributions.
- The variance of the classification accuracies of CONU is much smaller than that of the compared methods, indicating that CONU is very stable.
- The strong performance of CONU is due to the milder distribution assumptions and the effectiveness of the risk correction function in mitigating overfitting problems.

On real-world benchmark datasets, we can observe that

- CONU achieves the best performance in all cases, further confirming its effectiveness.
- The superiority is even more evident on CLCIFAR-20, a more complex dataset with extremely limited supervision. It demonstrates the advantages of CONU in dealing with real-world datasets.
- The performance of many state-of-the-art partial-label learning methods degenerates strongly, and many methods did not even work on CLCIFAR-20. The experimental results reveal their shortcomings in handling real-world data.

### J.3 EXPERIMENTAL RESULTS ON SYNTHETIC INSTANCE-DEPENDENT DATASETS

We followed Anonymous (2023) to generate synthetic complementary labels. First, we trained a neural network on ordinary labels to generate labeling confidence for each instance. Then, we sample a complementary label from the labels with the smallest $K$ labeling confidence. Then, we train a new model on the generated CL data. We used ResNet for CIFAR-10 and an MLP for other datasets as the backbone model. The hyperparameters are the same as before. Table 7 shows the experimental results with synthetic instance-dependent complementary labels. We found that our proposed method is still superior in this setting, which further validates the effectiveness of our method.

Table 7: Classification accuracy (mean±std) of each method with synthetic instance-dependent complementary labels. The best performance is shown in bold.

| Dataset | MNIST | | Kuzushiji-MNIST | | Fashion-MNIST | | CIFAR-10 | |
|---|---|---|---|---|---|---|---|---|
| # Min | 5 | 3 | 5 | 3 | 5 | 3 | 7 | 5 |
| SCL-NL | 61.01 ± 2.04 | 37.22 ±0.49 | 53.13 ±3.18 | 34.06 ±3.95 | 33.99 ±18.8 | 19.99 ±12.7 | 28.81 ±1.79 | 26.76 ±0.14 |
| SCL-EXP | 46.68 ±4.98 | 29.64 ±0.28 | 49.00 ±1.59 | 31.63 ±3.82 | 25.81 ±18.4 | 16.23 ±8.30 | 27.07 ±0.27 | 25.53 ±1.04 |
| NN | 49.34 ±1.87 | 22.67 ±2.00 | 38.54 ±2.97 | 26.59 ±2.23 | 40.42 ±6.87 | 25.66 ±5.04 | 38.28 ±0.81 | 31.68 ±1.60 |
| L-W | 35.29 ±2.29 | 25.45 ±0.98 | 32.59 ±0.56 | 25.97 ±1.56 | 21.52 ±5.98 | 17.58 ±2.54 | 29.59 ±1.00 | 26.27 ±0.80 |
| L-UW | 35.42 ±1.83 | 26.32 ±0.52 | 33.73 ±0.37 | 25.06 ±1.41 | 21.86 ±7.60 | 17.31 ±3.24 | 29.54 ±1.24 | 27.00 ±0.96 |
| GA | 44.26 ±1.75 | 29.26 ±1.91 | 39.75 ±2.30 | 24.20 ±0.59 | 30.50 ±13.2 | 11.54 ±1.47 | 37.88 ± 1.61 | 26.96 ±2.76 |
| Forward | 64.86 ±4.96 | 37.16 ±0.52 | 53.27 ±3.28 | 34.26 ±3.80 | 33.74 ±18.7 | 21.11 ±11.8 | 29.01 ±1.27 | 26.43 ±0.68 |
| CONU | **65.60** ±**0.94** | **46.38** ±**2.81** | **54.60** ±**1.92** | **43.40** ±**1.36** | **62.18** ±**10.5** | **42.75** ±**19.2** | **42.52** ±**0.92** | **35.60** ±**0.66** |

### J.4 EXPERIMENTAL RESULTS OF THE CLASS-PRIOR ESTIMATION APPROACH

We assumed that the ground-truth class priors for all labels and datasets are 0.1, which means that the test set was balanced. We generated complementary labels using the SCAR assumption with $\bar{\pi}_k = 0.5$. We repeated the generation process with different random seeds for 5 times. Table 8 shows the experimental results of the proposed class-prior estimation approach in Appendix A. We can observe that the class priors are accurately estimated in general with the proposed class-prior estimation method.

### J.5 COMPARISONS WITH FORWARD ON THE CLCIFAR DATASETS

Since there are three complementary labels for each example, following Wang et al. (2023), we used them separately to generate single complementary-label datasets. Since the transition matrix for Forward is unknown, we assume it to be a uniform distribution for a fair comparison. Table 9 shows the experimental results of Forward and CONU on CLCIFAR. We can observe that CONU performs better than Forward on the CLCIFAR datasets.

Table 8: Estimated values (mean±std) of class priors.

| Label Index | 1 | 2 | 3 | 4 | 5 |
|---|---|---|---|---|---|
| MNIST | $0.104 \pm 0.011$ | $0.119 \pm 0.012$ | $0.110\pm0.009$ | $0.099\pm0.008$ | $0.101\pm0.010$ |
| Kuzushiji-MNIST | $0.108\pm0.026$ | $0.098\pm0.011$ | $0.087\pm0.012$ | $0.104\pm0.004$ | $0.101\pm0.021$ |
| Fashion-MNIST | $0.091\pm0.016$ | $0.118\pm0.005$ | $0.090\pm0.024$ | $0.090\pm0.009$ | $0.077\pm0.020$ |
| CIFAR-10 | $0.085\pm0.016$ | $0.102\pm0.039$ | $0.073\pm0.019$ | $0.109\pm0.047$ | $0.100\pm0.031$ |

| Label Index | 6 | 7 | 8 | 9 | 10 |
|---|---|---|---|---|---|
| MNIST | $0.087\pm0.007$ | $0.089\pm0.005$ | $0.106\pm0.019$ | $0.091\pm0.008$ | $0.096\pm0.016$ |
| Kuzushiji-MNIST | $0.095\pm0.010$ | $0.105\pm0.025$ | $0.095\pm0.007$ | $0.094\pm0.016$ | $0.113\pm0.035$ |
| Fashion-MNIST | $0.117\pm0.007$ | $0.070\pm0.010$ | $0.114\pm0.023$ | $0.117\pm0.016$ | $0.117\pm0.016$ |
| CIFAR-10 | $0.098\pm0.013$ | $0.115\pm0.023$ | $0.120\pm0.033$ | $0.097\pm0.041$ | $0.100\pm0.013$ |

Table 9: Experimental results of Forward and CONU on CLCIFAR-10 and CLCIFAR-20 with single complementary labels. The CL Index indicates the used index of complementary labels.

| Dataset | CL Index | Model | Forward | CONU |
|---|---|---|---|---|
| CLCIFAR-10 | 1 | ResNet | 16.71 | 33.42 |
| CLCIFAR-10 | 1 | DenseNet | 17.70 | 37.53 |
| CLCIFAR-10 | 2 | ResNet | 16.37 | 32.27 |
| CLCIFAR-10 | 2 | DenseNet | 16.67 | 36.62 |
| CLCIFAR-10 | 3 | ResNet | 16.58 | 34.84 |
| CLCIFAR-10 | 3 | DenseNet | 17.15 | 36.41 |
| CLCIFAR-20 | 1 | ResNet | 5.00 | 13.34 |
| CLCIFAR-20 | 1 | DenseNet | 5.00 | 13.94 |
| CLCIFAR-20 | 2 | ResNet | 5.00 | 13.22 |
| CLCIFAR-20 | 2 | DenseNet | 5.00 | 13.24 |
| CLCIFAR-20 | 3 | ResNet | 5.00 | 12.04 |
| CLCIFAR-20 | 3 | DenseNet | 5.00 | 10.91 |

