# OpenReview forum: "Learning with Complementary Labels Revisited: A Consistent Approach via Negative-Unlabeled Learning"
_ICLR.cc/2024/Conference — Submitted to ICLR 2024_

### Official Review · Reviewer_CHdE · 2023-10-30

**Soundness:** 3 good
**Presentation:** 3 good
**Contribution:** 3 good
**Rating:** 6
**Confidence:** 4

**Summary:**

This paper addresses complementary-label learning, which is a weakly supervised learning problem. The proposed method does not rely on the uniform distribution assumption nor on the ordinary-label training set. More importantly, this method is risk-consistent with theoretical guarantees. Experiments on both synthetic and real-world benchmark datasets validate the effectiveness of the proposed approach.

**Strengths:**

1. The proposed method does not rely on the uniform distribution assumption nor an ordinary-label training set, which is more realistic.
2. The proposed method is risk-consistent with solid theoretical guarantees.
3. The introduction part is well-written and easy to follow.

**Weaknesses:**

1. The notations are confused, e.g., $\overline{Y}$ is a set of label vectors composed of {0,1}, and $k$ denotes the number of classes in {1,2, ..., q}, however, on page 4, line 5, the authors claim that "$k\in\overline{Y} $". The notations should be carefully checked and standardized.
2. Although the experimental results of the proposed method in the paper are better than the compared methods, there is a lack of experimental analysis provided. A thorough analysis of the experimental results would be helpful in understanding the factors contributing to the superior performance.
3. Although this paper provides nice theoretical results, it does not explain why the proposed method performs well. Which part plays an important role, the one-versus-rest (OVR) strategy, the risk correction function, or any other techniques?
4. The details of the constant $c_k$ in assumption 1 are missed. How to decide this constant? Will the choice of this constant affect the performance?

**Questions:**

1. Why set the label $\bar{y}$ as a q-dimensional vector? Intuitively, adding more complementary labels would improve the performance.
2. The assumptions in Theorem 2 directly borrow from Theorem 10 of [1]. According to these assumptions, which functions can be used as the loss function $l$? Can you list some of these and write the explicit form of $l$ that can be used in practice?
3. The compared method GA[2] also adopts a risk correction approach, can you explain why your CONU outperforms GA?

[1] Tong Zhang. Statistical analysis of some multi-category large margin classification methods. JMLR, 2004.

[2] Takashi Ishida, Gang Niu, Aditya K. Menon, and Masashi Sugiyama. Complementary-label learning for arbitrary losses and models. ICML 2019.

---

> ### Author Response · Authors · 2023-11-17
> **Response To Reviewer CHdE (1/2)**
>
> First, we are very grateful for your time and effort in reviewing this paper. Below are the responses to your questions and comments.
>
> **C1: The notations are confused. $\bar{Y}$ is the set of label vectors but the authors said $k\in\bar{Y}$.**
>
> **A1:** Sorry for the confusion, but the notation in the paper is correct. The paper uses two notations to denote the complementary label set for each example, $\bar{Y}$ and $\bar{\boldsymbol{y}}$. Here, $\bar{Y} \subseteq \mathcal{Y}$ is a **set of labels** which is a subset of all the labels in the label space $\mathcal{Y}$, not a set of label vectors. Besides, $\bar{\boldsymbol{y}}=\left[\bar{y}\_{1},\bar{y}\_{2},\ldots,\bar{y}\_{q}\right] \in \\{0, 1\\}^q$ is a single label vector that also denotes the complementary labels of this example, where $\bar{y}\_{k}=1$ if $k \in \bar{Y}$ and $\bar{y}\_{k}=0$ otherwise. We refer to the notation used in multi-label classification, where each example is associated with multiple true labels [1].
>
> **C2: A thorough analysis of the experimental results would be helpful.**
>
> **A2:** Yes, adding experimental analysis is very helpful. We have added a detailed analysis of the experimental results in Appendix J in the updated version of our paper, and we list them here.
>
> On synthetic benchmark datasets, we can observe
> - Out of 40 cases of different distributions and datasets, CONU achieves the best performance in 39 cases, which clearly validates the effectiveness of the proposed approach.
> - Some consistent approaches based on the uniform distribution assumption can achieve comparable or better or comparable performance to CONU for the "uniform" setting. For example, GA outperforms CONU on CIFAR-10. However, its performance drops significantly on other distribution settings. This shows that methods based on the uniform distribution assumption cannot handle more realistic data distributions.
> - The variance of the classification accuracies of CONU is much smaller than that of the compared methods, indicating that CONU is very stable.
> - The strong performance of CONU is due to the milder distribution assumptions and the effectiveness of the risk correction function in mitigating overfitting problems.
>
> On real-world benchmark datasets, we can observe that
> - CONU achieves the best performance in all cases, further confirming its effectiveness.
> - The superiority is even more evident on CLCIFAR-20, a more complex dataset with extremely limited supervision. It demonstrates the advantages of CONU in dealing with real-world datasets.
> - The performance of many state-of-the-art partial-label learning methods degenerates strongly, and many methods did not even work on CLCIFAR-20. The experimental results reveal their shortcomings in handling real-world data.
>
> **C3: Why the proposed method performs well?**
>
> **A3:** It is exciting to observe the strong performance of our method because we did not use any strong regularization techniques or even tune the hyperparameters. We believe that the effectiveness is due to: 1) CONU is based on a milder data distribution assumption. We do not rely on the uniform data distribution or an ordinary-label dataset. Therefore, our method can handle more complex data distributions in more realistic scenarios. 2) The risk correction function is very effective in mitigating overfitting problems. From Figure 1, we can see that the loss function without the risk correction function encounters severe overfitting problems. However, with the help of the risk correction function, the performance is greatly improved.
>
> **C4: How to decide $c_k$ and what is its influence?**
>
> **A4:** Thanks for your question. We have added in Assumption 1 that $c_k=\bar{\pi}\_{k}/(1-\pi_k)$. Therefore, $c_k$ represents the fraction of complementary-label data with the $k$-th class as a complementary label in all the examples not belonging to the $k$-th class. The value of $c_k$ is determined by how many complementary labels are annotated. If more complementary labels are given by the annotator, $c_k$ will be larger. Then, the performance of complementary-label learning methods will be better since more supervision information is given.
>
> **C5: Why set the label $\bar{\boldsymbol{y}}$ as a $q$-dimensional vector? Intuitively, adding more complementary labels would improve the performance.**
>
> **A5:** We use this symbol only for ease of notation, since we need the marginal densities where the $k$-th class is considered as complementary or not in our formulation, i.e. $p\left(\boldsymbol{x}|\bar{y}\_{k}=1\right)$ and $p\left(\boldsymbol{x}|\bar{y}\_k=0\right)$. We believe this notation is clearer. Also, of course, adding more complementary labels can improve performance. If we add more complementary labels, more elements in $\bar{\boldsymbol{y}}$ will change from 0 to 1.

---

> ### Author Response · Authors · 2023-11-17
> **Response To Reviewer CHdE (2/2)**
>
> **C6: Which functions can be used as the loss function $\ell$?**
>
> **A6:** Any binary-class loss function $\ell$ that is convex, bounded below, differential, and satisfies $\ell(z)\leq \ell(-z)$ when $z>0$ can make Theorem 2 hold. We list a few most common loss functions below:
> |Name|$\ell(z)$|
> |---|---|
> |Squared|$(1-z)^2$|
> |Logistic|$\log(1+\exp(-z))$|
> |Exponential|$\exp(-z)$|
>
> **C7: Can you explain why your CONU outperforms GA?**
>
> **A7:** We can see that on CIFAR-10 with the uniform distribution, GA outperforms CONU. This shows that we cannot rely on one approach for all datasets, and each approach has its own advantage. However, on other distribution settings, CONU outperforms GA because GA relies on the uniform distribution assumption while CONU relies on a milder assumption. The performance of GA degenerates significantly when the complementary labels do not follow the uniform distribution.
>
> ***
> Reference:
>
> [1] Collaborative learning of label semantics and deep label-specific features for multi-label classification, TPAMI 2022.

---

> > ### Comment · Reviewer_CHdE · 2023-11-22
> >
> > Thanks for the responses. I have a better understanding of the paper now. According to the explanation of the authors, the competitive performance mainly comes from the milder data distribution assumption.
> >
> > On page 4 line 4, the authors claim that "use a q-dimensional label vector $\bar{y} = [\bar{y}_1, \bar{y}_2, \cdots, \bar{y}_q ]  \in$ {0,1}$^q$ to denote $\bar{Y}$".  However, $ k $ denotes the number of classes in {1,2, ..., q}, thus, it is not correct to say that $k \in \bar{Y}$. I understand what the authors mean, but math should be rigorous and correct.
> >
> > I will make my final score after the discussion with other reviewers, and I currently retain my score.

---

> > > ### Author Response · Authors · 2023-11-22
> > >
> > > Thanks for furthering the discussion. In our notation system, $\bar{Y}$ denotes the set of complementary labels for an example, while $\bar{\boldsymbol{y}}$ is its vector version, defined as $\bar{\boldsymbol{y}}=\left[\bar{y}\_{1},\bar{y}\_{2},\ldots,\bar{y}\_{q}\right] \in \\{0, 1\\}^q$. For example, in a 4-class classification problem, if $\bar{Y} = \\{1,2,4\\}$, then the corresponding $\bar{\boldsymbol{y}}$ is [1,1,0,1]. We apologize for the previous inaccurate description of these notations, and have rewritten this sentence to make it clearer.
> > >
> > > Thank you for your help and time with this submission.

---

> ### Author Response · Authors · 2023-11-20
>
> Dear Reviewer CHdE,
>
> We sincerely appreciate your time and effort in reviewing our paper. As the discussion period between reviewers and authors will end in two days, if you have any further concerns or questions, please do not hesitate to raise them. Thank you very much!
>
> Best,
>
> ICLR 2024 Conference Submission2453 Authors

---

### Official Review · Reviewer_XP12 · 2023-10-31

**Soundness:** 2 fair
**Presentation:** 2 fair
**Contribution:** 2 fair
**Rating:** 3
**Confidence:** 3

**Summary:**

This manuscript utilized the OVR strategy to decompose the complementary-label learning into a set of negative-unlabeled classification problems.

**Strengths:**

This manuscript utilized the OVR strategy to decompose the complementary-label learning into a set of negative-unlabeled classification problems.

**Weaknesses:**

1. ``Existing consistent approaches have relied on the uniform distribution assumption to model the generation of complementary labels, or on an ordinary-label training set to estimate the transition matrix. '' This argument is false because some cll algorithms are designed for both uniform and non-uniform distribution.

2. The methodologies compared in this research are outdated, if not obsolete. I'm afraid that most cutting-edge techniques are missing.

3. Lack of comparison. Because cll is a subset of pll, the methods comparison should include the most recent pll methods.

4. The loss proposed for complementary label learning is uncear.

5. The experiments are conducted on simple datasets, some complex dataset like cifar100, subset of webvision should be used to verify the effectness of the proposed method.

**Questions:**

1. Is this strategy of this paper is similar to a previous work of Ishida2017[1].

[1] Takashi Ishida, Gang Niu, Weihua Hu, and Masashi Sugiyama. Learning from complementary
labels. In Advances in Neural Information Processing Systems 30, pp. 5644–5654, 2017.

---

> ### Author Response · Authors · 2023-11-17
> **Clarifications Of Misunderstandings In The Comments Made By Reviewer XP12 (1/2)**
>
> Thank you for reviewing our paper. There are **misunderstandings** of our paper in your review, including **several factual errors and biased opinions**. Therefore, please allow us to respectfully correct your misunderstandings of our paper. Below are our responses to your comments.
>
> **C1: The following argument is false.**
> > **Existing consistent complementary-label learning (CLL) approaches have relied on the uniform distribution assumption or on an ordinary-label training set to estimate the transition matrix.**
>
> **A1:** We respectfully **disagree** with your comment. Your reason that "some CLL algorithms are designed for both uniform and non-uniform distributions" is one thing, while our argument that "consistent biased-distribution methods need to use an ordinary-label training set to estimate the transition matrix" is another thing. So your reason is **not at all relevant** to our argument, let alone proving our argument wrong. We have written very clearly that the focus of this paper is on **consistent** CLL approaches, all of which are listed in Table 1. It is very clear that our argument that "existing consistent CLL approaches all require the uniform distribution assumption or an ordinary-label training set to estimate the transition matrix" is **correct**. You just used another fact A to deny our argument B, which is completely inappropriate.
>
> Also, **all three other reviewers agreed with our argument**. Reviewer MvNX said that "the proposed approach doesn’t rely on assumptions about the distribution of complementary labels or ordinary-label training set, which makes it more suitable for real-world scenarios". Reviewer ye4F said that "method that avoiding the assumptions on the transition matrix is a substantial contribution to the CLL community". Reviewer CHdE said that "the proposed method does not rely on the uniform distribution assumption nor an ordinary-label training set, which is more realistic". Therefore, they all consider this argument to be correct, which is contrary to your comment. Since this argument is the main motivation of our approach, we are afraid that such factual misunderstanding leads directly to your biased evaluation of our paper.
>
> **C2: Lack of comparison. The methods comparison should include the most recent pll methods.**
>
> **A2:** We still respectfully **disagree** with your comment. We have included **14 methods compared** in this paper, of which **8** are PLL methods. We also included a very recent SOTA PLL approach, IDGP [1]. Therefore, the comparison is **sufficient** to validate the effectiveness of our method. Also, we would like to point out that it is unfair to compare our method to some PLL approaches with strong regularization loss functions and data augmentation techniques, since we use only a single loss function without any strong regularization techniques. Nevertheless, as you suggest, we have included two SOTA PLL approaches proposed in ICML 2023 [2], namely PRODEN-POP and CAVL-POP. Since they have outperformed almost all other PLL methods according to their respective literature, we only present the performance of them. Here are the experimental results against these three recent SOTA PLL approaches on the two real-world benchmark datasets:
>
> |Dataset|Model|IDGP|PRODEN-POP|CAVL-POP|CONU|
> |---|---|---|---|---|---|
> |CLCIFAR-10|ResNet|10.00|25.68|10.00|**42.04**|
> |CLCIFAR-10|DenseNet|10.00|30.66|10.00|**44.41**|
> |CLCIFAR-20|ResNet|4.96|6.73|5.00|**20.08**
> |CLCIFAR-20|DenseNet|5.00|5.00|5.00|**19.91**
>
>
> We can see that our proposed approach performs better than them, which still validates its effectiveness.
>
> **C3: The methodologies compared in this research are outdated. I'm afraid that most cutting-edge techniques are missing.**
>
> **A3:** We included the most recent CLL and PLL approaches in 2023, i.e., OP [3] and IDGP [1], respectively, so we are very confused about what the "most cutting-edge techniques" are.
>
> **C4: The loss proposed for complementary label learning is unclear.**
>
> **A4:** On page 6, line 4, we said
> > We refer to the method that works by minimizing the corrected risk estimator in Eq. (9) introduced below as CONU, where the algorithm details are summarized in Appendix B.
>
> In the first two lines in Appendix H.3, we said
> > The logistic loss was adopted to instantiate the binary-class loss function $\ell$, and the absolute value function was used as the risk correction function $g$ for CONU.
>
> Therefore, the proposed loss is **very clear** in this paper.

---

> ### Author Response · Authors · 2023-11-17
> **Clarifications Of Misunderstandings In The Comments Made By Reviewer XP12 (2/2)**
>
> **C5: we should include experiments on CIFAR-100.**
>
> **A5:** Contrary to your comment, we did include experiments on a variant of CIFAR-100 in the paper. We used CLCIFAR-20, a recent CLL benchmark based on CIFAR-100 with superclass labels. Therefore, **this comment is also unreasonable**.
>
> **C6: We should include experiments on WebVision.**
>
> **A6:** We still respectfully **disagree** with this comment. We didn't use WebVision because it is a noisy-label learning dataset, we cannot ensure that the complementary label generated is not the true label. Thus, no previous work in PLL or CLL takes this into account, and we simply followed their protocol. We would appreciate it if you could **provide a reference** in the CLL literature that uses this dataset to support your comment.
>
> **C7: The strategy is similar to OVA [4].**
>
> **A7:** Please allow me to list these two loss functions.
>
> The loss function of OVA [4]:
> \begin{equation}
> \widehat{R}(f)=\frac{K-1}{n} \sum_{i=1}^n \overline{\mathcal{L}}(f(\boldsymbol{x}\_i), \bar{y}\_i)-M_1+M_2,
> \end{equation}
> where
> \begin{equation}
> \overline{\mathcal{L}}(f(\boldsymbol{x}), \bar{y})=\frac{1}{K-1} \sum_{y \neq \bar{y}} \ell\left(g_y(\boldsymbol{x})\right)+\ell\left(-g_{\bar{y}}(\boldsymbol{x})\right).
> \end{equation}
> The loss function of CONU:
> \begin{equation}
> \widetilde{R}\left(f_{1}, f_{2}, \ldots, f_{q}\right)=\sum_{k=1}^{q}\widetilde{R}\_{k}(f_{k}),
> \end{equation}
> where
> \begin{equation}
> \widetilde{R}\_{k}(f_{k})=g\left(\widehat{R}^{\rm P}\_{k}(f\_{k})\right)+\frac{1-\pi_{k}}{n^{\rm N}\_{k}}\sum_{i=1}^{n^{\rm N}\_{k}}\ell\left(-f_{k}(\boldsymbol{x}\_{k,i}^{\mathrm{N}})\right)
> \end{equation}
> and
> \begin{equation}
> \widehat{R}^{\rm P}\_{k}(f_{k})=\frac{\bar{\pi}\_{k}+\pi_{k}-1}{n^{\rm N}\_{k}}\sum_{i=1}^{n^{\rm N}\_{k}}\ell\left(f_{k}(\boldsymbol{x}\_{k,i}^{\mathrm{N}})\right)+\frac{1-\bar{\pi}\_{k}}{n^{\rm U}\_{k}}\sum\_{i=1}^{n^{\rm U}\_{k}}\ell\left(f_{k}(\boldsymbol{x}\_{k,i}^{\mathrm{U}})\right).
> \end{equation}
> It is very clear that the two losses are completely different, even though they can both use the logistic loss to instantiate $\ell$. Apart from the loss function, our differences are: 1) OVA is only designed for learning with single complementary labels, while CONU can handle any number of complementary labels; 2) The data distribution assumptions are different. OVA is based on the uniform distribution assumption, while CONU is based on the SCAR assumption. Therefore, our method is not similar to OVA at all.
>
> In summary, there are many factual misunderstandings and biased opinions in your review. We hope that this rebuttal corrects your misunderstandings. We look forward to discussing your comments with you in this discussion period.
> ***
> Reference:
>
> [1] Decompositional generation process for instance-dependent partial label learning, ICLR 2023.
>
> [2] Progressive purification for instance-dependent partial label learning, ICML 2023.
>
> [3] Consistent complementary-label learning via order-preserving losses, AISTATS 2023.
>
> [4] Learning from complementary labels, NeurIPS 2017.

---

> ### Author Response · Authors · 2023-11-20
> **Need further clarification?**
>
> Dear Reviewer XP12,
>
> Thank you very much for reviewing our paper. We have tried our best to clarify the misunderstandings and factual errors in your comments. **We believe that our responses can correct your misunderstandings of our paper**. Since the discussion period between authors and reviewers **ends after two days**, please let us know if you **need any further clarification**.
>
>
> Best,
>
> ICLR 2024 Conference Submission2453 Authors

---

### Official Review · Reviewer_ye4F · 2023-10-31

**Soundness:** 4 excellent
**Presentation:** 3 good
**Contribution:** 3 good
**Rating:** 6
**Confidence:** 3

**Summary:**

This paper tackles complementary label learning by regarding the problem as multiple negative-unlabeled learning problems. This novel formulation avoids explicit assumptions on the label distribution relationship between complementary and ground-truth labels, and is risk-consistent with theoretical guarantees: A risk-consistent estimator. Empirical results validate promising performance of the proposed approach.

**Strengths:**

1. Solid justifications on the statistical consistency and convergence rate of the corrected risk estimator have been provided with proofs.
2. Method that avoiding the assumptions on the transition matrix is a substantial contribution to the CLL community.

**Weaknesses:**

1. More experiments with instance-dependent CL data should be investigated, due to the practical reason mentioned in this paper.
2. The performance of prior estimation should be evaluated in the empirical study.

**Questions:**

1. I notice some results of existing methods are much worse than the results reported in their original papers. For example, the results of NN and GA on K-MINST and F-MNIST in paper [1] are much higher than that are reported in your paper.
2. Have you tried FORWARD [2] on CLCIFAR datasets? I notice that the results of FORWAD is pretty good on these instance-dependent CL datasets and should be involved in the comparison.

Refs.\
[1] Chou Y T, Niu G, Lin H T, et al. Unbiased risk estimators can mislead: A case study of learning with complementary labels[C]//International Conference on Machine Learning. PMLR, 2020: 1929-1938.\
[2] Xiyu Yu, Tongliang Liu, Mingming Gong, and Dacheng Tao. Learning with biased complementary labels. In Proceedings of the 15th European Conference on Computer Vision, pp. 68–83, 2018.

---

> ### Author Response · Authors · 2023-11-17
> **Response To Reviewer ye4F (1/2)**
>
> First, we would like to thank you for your time and effort in reviewing our submission. We are very encouraged that you agree with our contributions. Below are the responses to your comments.
>
> **C1: More experiments with instance-dependent CL data should be investigated.**
>
> **A1:** We have added experiments on synthetic instance-dependent CL data. We noticed that there is another ICLR 2024 submission [1] working on instance-dependent CLL. Therefore, we followed their procedures to generate complementary labels. First, we trained a neural network on ordinary labels to generate labeling confidence for each instance. Then, we sample a complementary label from the labels with the smallest $K$ labeling confidence. Then, we train a new model on the generated CL data. We used ResNet for CIFAR-10 and an MLP for other datasets as the backbone model. The hyperparameters are the same as in our paper. Here are the experimental results, which are also presented in Appendix J of the updated version of the paper:
>
> |Dataset|\#Min|SCL-NL|SCL-EXP|NN|L-W|L-UW|GA|Forward|CONU|
> |---|---|---|---|---|---|---|---|---|---|
> |MNIST|5|61.01 ± 2.04|46.68 ± 4.98|49.34 ± 1.87|35.29 ± 2.29|35.42 ± 1.83|44.26 ± 1.75|64.86 ± 4.96|**65.60 ± 0.94**|
> |MNIST|3|37.22 ± 0.49|29.64 ± 0.28|22.67 ± 2.00|25.45 ± 0.98|26.32 ± 0.52|29.26 ± 1.91|37.16 ± 0.52|**46.38 ± 2.81**|
> |Kuzushiji-MNIST|5|53.13 ± 3.18|49.00 ± 1.59|38.54 ± 2.97|32.59 ± 0.56|33.73 ± 0.37|39.75 ± 2.30|53.27 ± 3.28|**54.60 ± 1.92**|
> |Kuzushiji-MNIST|3|34.06 ± 3.95|31.63 ± 3.82|26.59 ± 2.23|25.97 ± 1.56|25.06 ± 1.41|24.20 ± 0.59|34.26 ± 3.80|**43.40 ± 1.36**|
> |Fashion-MNIST|5|33.99 ± 18.8|25.81 ± 18.4|40.42 ± 6.87|21.52 ± 5.98|21.86 ± 7.60|30.50 ± 13.2|33.74 ± 18.7|**62.18 ± 10.5**|
> |Fashion-MNIST|3|19.99 ± 12.7|16.23 ± 8.30|25.66 ± 5.04|17.58 ± 2.54|17.31 ± 3.24|11.54 ± 1.47|21.11 ± 11.8|**42.75 ± 19.2**|
> |CIFAR-10|7|28.81 ± 1.79|27.07 ± 0.27|38.28 ± 0.81|29.59 ± 1.00|29.54 ± 1.24|37.88 ± 1.61|29.01 ± 1.27|**42.52 ± 0.92**|
> |CIFAR-10|5|26.76 ± 0.14|25.53 ± 1.04|31.68 ± 1.60|26.27 ± 0.80|27.00 ± 0.96|26.96 ± 2.76|26.43 ± 0.68|**35.60 ± 0.66**|
>
> We found that our proposed method is still superior in this setting, which further validates the effectiveness of our method.
>
> **C2: The performance of prior estimation should be evaluated in the empirical study.**
>
> **A2:** Thank you for your suggestion. We have presented the experimental results of the proposed class prior estimation approach in Appendix A. We assumed that the ground-truth class priors for all labels and datasets are 0.1, which means that the test set was balanced. We generated complementary labels using the SCAR assumption with $\bar{\pi}\_{k}=0.5$. Here are the experimental results:
>
> |Label Index| 1| 2| 3| 4| 5| 6| 7| 8| 9| 10|
> |---|---|---|---|---|---|---|---|---|---|---|
> |MNIST|0.104±0.011|0.119±0.012|0.110±0.009|0.099±0.008|0.101±0.010|0.087±0.007|0.089±0.005|0.106±0.019|0.091±0.008|0.096±0.016|
> |Kuzushiji-MNIST|0.108±0.026|0.098±0.011|0.087±0.012|0.104±0.004|0.101±0.021|0.095±0.010|0.105±0.025|0.095±0.007|0.094±0.016|0.113±0.035|
> |Fashion-MNIST|0.091±0.016|0.118±0.005|0.090±0.024|0.090±0.009|0.077±0.020|0.117±0.007|0.070±0.010|0.114±0.023|0.117±0.016|0.117±0.016|
> |CIFAR-10|0.085±0.016|0.102±0.039|0.073±0.019|0.109±0.047|0.100±0.031|0.098±0.013|0.115±0.023|0.120±0.033|0.097±0.041|0.100±0.013|
>
>
> We can observe that the class priors are accurately estimated in general with the proposed method.
>
> **C3: Some results of existing methods are much worse than the results reported in their original papers.**
>
> **A3:** The different experimental results are due to different experimental settings. The main difference is in the hyperparameters. We believe that it is not trivial to tune hyperparameters in complementary-label learning without an ordinary-label dataset. A common practice in machine learning is to separate the validation set from the training set. However, since the ground-truth labels of complementary-label data are unknown, it is difficult to use a complementary-label validation set to tune hyperparameters. Therefore, for a fair comparison, we did not tune specific hyperparameters and fixed the hyperparameters for all compared methods to the same values. For example, the learning rate was set to 1e-3. Details of the hyperparameters can be found in Appendix H. In paper [2], the authors said "we used the Adam optimizer with a learning rate selected from {1e-1, 1e-2, 1e-3, 1e-4, 1e-5} and trained the models for 300 epochs". So we assumed that they used an ordinary-label dataset to tune the hyperparameters, which is why their results were better.

---

> ### Author Response · Authors · 2023-11-17
> **Response To Reviewer ye4F (2/2)**
>
> **C4: The results of Forward is pretty good on these instance-dependent CL datasets and should be involved in the comparison.**
>
> **A4:** As you suggested, we have added the experimental results of Forward on instance-dependent CL datasets. In A1, we have added experimental results of Forward on synthetic instance-dependent CL datasets. The comparison with CONU on CLCIFAR-10 and CLCIFAR-20 is done here. Since there are three complementary labels for each example, following [3], we used them separately to generate single complementary-label datasets. Since the transition matrix for Forward is unknown, we assume it to be a uniform distribution for a fair comparison. Here are the experimental results:
>
> |Dataset|CL Index|Model|Forward|CONU|
> |---|---|---|---|---|
> |CLCIFAR-10|1|ResNet|16.71|**33.42**|
> |CLCIFAR-10|1|DenseNet|17.70|**37.53**|
> |CLCIFAR-10|2|ResNet|16.37|**32.27**|
> |CLCIFAR-10|2|DenseNet|16.67|**36.62**|
> |CLCIFAR-10|3|ResNet|16.58|**34.84**|
> |CLCIFAR-10|3|DenseNet|17.15|**36.41**|
> |CLCIFAR-20|1|ResNet|5.00|**13.34**|
> |CLCIFAR-20|1|DenseNet|5.00|**13.94**|
> |CLCIFAR-20|2|ResNet|5.00|**13.22**|
> |CLCIFAR-20|2|DenseNet|5.00|**13.24**|
> |CLCIFAR-20|3|ResNet|5.00|**12.04**|
> |CLCIFAR-20|3|DenseNet|5.00|**10.91**|
>
> We can observe that CONU performs better than Forward on the CLCIFAR datasets.
>
> ***
> Reference:
>
> [1] Rethinking one-vs-the-rest loss for instance-dependent complementary label learning, ICLR 2024 submission.
>
> [2] Unbiased risk estimators can mislead: A case study of learning with complementary labels, ICML 2020.
>
> [3] CLCIFAR: CIFAR-derived benchmark datasets with human annotated complementary labels, arXiv 2023.

---

> > ### Comment · Reviewer_ye4F · 2023-11-22
> >
> > Thanks for your response. However, the result of FORWARD on CLCIFAR10 is too bad according to your results, which is not consistent to my own experimental results. My final decision will be released soon, after the discussion with the AC and other reviewer. Thanks.

---

> ### Author Response · Authors · 2023-11-20
>
> Dear Reviewer ye4F,
>
> We sincerely appreciate your time and effort in reviewing our paper. As the discussion period between reviewers and authors will end in two days, if you have any further concerns or questions, please do not hesitate to raise them. Thank you very much!
>
> Best,
>
> ICLR 2024 Conference Submission2453 Authors

---

> ### Author Response · Authors · 2023-11-22
> **Our Experimental Results Are Reproducible And Fair**
>
> Thank you for your response. We would like to clarify that our experimental results are reproducible, valid and fair.
>
> **First of all, our results are absolutely reproducible**.
>
> We have included the experimental codes of CLCIFAR-10 and CLCIFAR-20 in the "Supplementary Material". You can run our code and get results similar to ours. Please put the CIFAR-10 (100) dataset, the ".pkl" files of CLCIFAR-10 and CLCIFAR-20 into "CLCIFAR/dataset" and use the following scripts:
> ```
> python main.py -ds clcifar10 -me Forward -mo resnet -op adam -lr 1e-3 -wd 1e-3 -bs 256 -ep 200 -seed 0
> python main.py -ds clcifar10 -me CONU -mo resnet -op adam -lr 1e-3 -wd 1e-3 -bs 256 -ep 200 -seed 0
> ```
>
> **Second, our experiments are fair and valid.**
>
> We did not use the code implementation of [1]. This is because they use a **clean validation set** with true labels for hyperparameter tuning in their code, which may not be realistic for CLL. **If we have a clean dataset, why not use it for training?** Since the supervision information in CLL is very limited, adding an ordinary-label training set can help a lot [2]. Besides, as a common sense in machine learning, the validation set is divided from the training set. However, we do not know the true labels of the training data. So, we argue that it may not be valid to use a clean validation set for hyperparameter tuning.
>
> We implement the code framework ourselves and **set the same hyperparameters (e.g. learning rates, weight decay)** for all methods. The specific hyperparameters can be seen from our code scripts. We believe that the comparison is **very fair**, although the performance of all methods will degenerate because the hyperparameters are not hand-picked for each method and each dataset. Of course, if we tune the hyperparameters for all methods with a clean validation set, the performance of all of them will be better. So in their setting [1], our method can also achieve better performance compared with our results.
>
> I hope this rebuttal addresses your concerns. If you have any further concerns or questions, please do not hesitate to ask. Thank you again for your time and effort in reviewing our submission.
>
> ***
> Reference:
>
> [1] CLCIFAR: CIFAR-derived benchmark datasets with human annotated complementary labels, arXiv 2023.
>
> [2] Learning from complementary labels, NeurIPS 2017.

---

### Official Review · Reviewer_MvNX · 2023-11-01

**Soundness:** 3 good
**Presentation:** 3 good
**Contribution:** 2 fair
**Rating:** 6
**Confidence:** 4

**Summary:**

This paper points out that the existing complementary-label learning approaches have relied on some assumptions about the distribution of complementary labels, or on an ordinary-label training set, which may not be satisfied in real-world scenarios. It then proposes a risk-consistent approach that express complementary-label learning as a set of negative-unlabeled binary classification problems, using the one-versus-rest strategy. Furthermore, it introduce a risk correction approach to address overfitting problems when using complex models. It also proves the statistical consistency and convergence rate of the corrected risk estimator.

**Strengths:**

1. The idea of expressing complementary-label learning as a set of negative-unlabeled binary classification problems is very novel and sensible.
2. The proposed approach doesn’t rely on assumptions about the distribution of complementary labels or ordinary-label training set, which makes it more suitable for real-world scenarios.
3. The result is promising.

**Weaknesses:**

1. The major novelty lies in the problem reformulation. The way to conduct theoretical analysis and risk correction is off-the-shelf. In this sense, the technical contribution is not very impressive.
2. In Fig2, only the impact of inaccurate class priors over the proposed method is illustrated. How about the competitors?
3. Some recent PU-learning methods should be reviewed in Sec.2.2, e.g.,
[1] Beyond Myopia: Learning from Positive and Unlabeled Data through Holistic Predictive Trends. NeurIPS 2023.

[2] Positive-Unlabeled Learning With Label Distribution Alignment. TPAMI 2023.

[3] GradPU: Positive-Unlabeled Learning via Gradient Penalty and Positive Upweighting. AAAI 2023.

[4] Dist-PU: Positive-Unlabeled Learning From a Label Distribution Perspective. CVPR 2022.

**Questions:**

A brief introduction for the compared methods to explain their respective characteristics will help understanding.

---

> ### Author Response · Authors · 2023-11-17
> **Response To Reviewer MvNX**
>
> First of all, we are very grateful for your time and effort in reviewing this submission. We are encouraged that you agree with the novelty of our paper. Below are the responses to your comments.
> ***
> **Q1: The way to conduct theoretical analysis and risk correction is off-the-shelf. In this sense, the technical contribution is not very impressive.**
>
> **A1:** We would like to clarify that the technical contribution and the focus of our paper are not only the theoretical results. The theoretical results are just auxiliaries to show the solid theoretical guarantees of our proposed methods, but not the main contributions. We have revised the contribution part of Introduction in the updated version of our paper to make it clearer.
>
> First, we think that our attempt to get rid of the uniform distribution assumption and ordinary-label training sets is a **substantial contribution** to the CLL community, as also commented by Reviewer ye4F. All consistent CLL approaches (listed in Table 1) rely on the uniform distribution assumption or an ordinary-label training set, which are not realistic despite their solid theoretical properties. Therefore, designing a more realistic CLL approach with theoretical guarantees can make a great contribution and bring new insights to the field.
>
> Second, another major contribution is that we provide **a new way to understand CLL**. We uncover the relationship between CLL and negative-unlabeled learning, which is quite novel and critical to the CLL literature. To our knowledge, this relationship has never been explored in the CLL literature. We believe that this finding may provide new directions and insights for solving CLL problems. For example, in Remark 1, we explain that besides minimizing $R_{k}(f_k)$, we can use **any other PU learning approach** to derive the binary classifier $f_{k}$ by swapping the positive class and the negative class. Therefore, we have not only proposed a single loss function, but a **general framework that incorporates PU learning approaches to solve CLL problems**. Such a framework can also bring new insights to this field.
>
> Therefore, we believe that our work is a significant contribution to the CLL community.
>
> **Q2: The influence of class priors on the competitors.**
>
> **A2:** CONU is the only CLL approach that uses class priors in the loss function. Therefore, the ground-truth class priors have no effect on the compared approaches.
>
> **Q3: Some recent PU learning methods should be reviewed.**
>
> **A3:** Thanks for your suggestion. We have added these references and provided a review of PU learning in Appendix I in the updated version of our paper.
>
> **Q4: A brief introduction for the compared methods will help understanding.**
>
> **A4:** We agree with your comments and we have included the introductions in Appendix H in the updated version of our paper.

---

> ### Author Response · Authors · 2023-11-20
>
> Dear Reviewer MvNX,
>
> We sincerely appreciate your time and effort in reviewing our paper. As the discussion period between reviewers and authors will end in two days, if you have any further concerns or questions, please do not hesitate to raise them. Thank you very much!
>
> Best,
>
> ICLR 2024 Conference Submission2453 Authors

---

### Author Response · Authors · 2023-11-18
**Summary Of Our Rebuttal**

First of all, we sincerely thank all reviewers for their great efforts in reviewing this submission and providing helpful and valuable comments. Based on your comments, **we have uploaded the updated version of our paper with all changes highlighted in blue**. The changes include

- Clearer presentation of our contributions. In Introduction, we highlight our main contributions in two ways. First, we propose the first consistent complementary-label learning approach without relying on the uniform distribution assumption or an additional ordinary-label dataset. Second, we uncover the relationship between complementary-label learning and negative-unlabeled learning, which provides a new perspective for understanding complementary-label learning.

- Additional Experimental Results. We have added more experimental results in Appendix J, including experimental results on synthetic instance-dependent complementary-label learning datasets, experimental results of the proposed class-prior estimation method, and comparisons with Forward on the CLCIFAR datasets. We also provide a detailed analysis of the experimental results.

- More description of the methodology. We have added the definition of $c_k$ in Assumption 1, and added descriptions of all the compared methods in Appendix H.

- A brief introduction to PU learning and more related references.

---

### Meta-Review · Area_Chair_tr6Q · 2023-12-08

**Metareview:**

This paper studies the learning from complementary label problem and proposes a consistent approach without relying on the uniform distribution assumption or an additional ordinary-label dataset. The theoretical results are motivated by the relationship between complementary-label learning and negative-unlabeled learning. The authors also provide an analysis of the consistency and convergence rate. The method is validated on both synthetic and real-world datasets.

Strengths: most of the reviewers agree that the idea is novel, and the results are promising.

Weaknesses: the major concern that remains after the rebuttal is the novelty of the theoretical analysis and the fair comparison with previous methods. In terms of the novelty, reviewers mentioned that the theoretical analysis is not really novel but is still useful for this reformulation. In terms of the fair comparison, the authors mentioned that they used the same hyperparameters for all the methods. This is concerning as it is natural that different methods may need different hyperparameters. A more fair way is to choose the best parameters for each method. This is a separate issue from whether to use a clean validation set.

**Justification For Why Not Higher Score:**

This is a borderline paper. Given the remaining issues raised by the reviewer after the rebuttal, this paper will benefit from another round of revision and should include more methodology details and comparison with prior work in the next version.

**Justification For Why Not Lower Score:**

N/A

---

### Decision · Program_Chairs · 2024-01-16

Reject